# BETTER WITH LESS: DATA-ACTIVE PRE-TRAINING OF GRAPH NEURAL NETWORKS

## ABSTRACT

Recently, pre-training on graph neural networks (GNNs) has become an active research area and is used to learn transferable knowledge for downstream tasks with unlabeled data. The success of graph pre-training models is often attributed to the massive amount of input data. In this paper, however, we identify the *curse of big data* phenomenon in graph pre-training: more training samples and graph datasets do not necessarily lead to better performance. Motivated by this observation, we propose a *better-with-less* framework for graph pre-training: few, but carefully chosen data are fed into a GNN model to enhance pre-training. This novel pre-training pipeline is called the data-active graph pre-training (APT) framework, and is composed of a graph selector and a pre-training model. The graph selector chooses the most representative and instructive data points based on the inherent properties of graphs as well as the *predictive uncertainty*. The proposed predictive uncertainty, as feedback from the pre-training model, measures the confidence level of the model to the data. When fed with the chosen data, on the other hand, the pre-training model grasps an initial understanding of the new, unseen data, and at the same time attempts to remember the knowledge learnt from the previous data. Therefore, the integration and interaction between these two components form a unified framework, in which graph pre-training is performed in a progressive way. Experiment results show that the proposed APT framework is able to obtain an efficient pre-training model with fewer training data and better downstream performance.

## 1 INTRODUCTION

Pre-training Graph Neural Networks (GNNs) shows the potential to be an attractive and competitive strategy for learning graph representations without costly labels. However, its transferability is guaranteed only if the pre-training datasets come from the same or similar domain as the downstream Hu et al. (2019; 2020b); You et al. (2020a;b); Hu et al. (2020c); Li et al. (2021); Lu et al. (2021); Sun et al. (2021). When we have no knowledge of the downstream, an encouraging yet largely unexplored research direction is pre-training GNNs on cross-domain data Qiu et al. (2020); Hafidi et al. (2020). Taking the graphs from multiple domains as the input, graph pre-training is able to learn the transferable structural patterns in graphs (when some semantic meanings are present), or to obtain the capability of discriminating these patterns.

With diverse and various cross-domain data, the success of a graph pre-training model is often attributed to the massive amount of unlabeled training data, a well-established fact for pre-training in computer vision Girshick et al. (2014); Donahue et al. (2014); He et al. (2020) and natural language processing Mikolov et al. (2013); Devlin et al. (2019). In view of this, contemporary research almost has no controversy on the following issue: *Is a massive amount of input data really necessary, or even beneficial, for pre-training GNNs?* However, two simple experiments regarding the number of training samples and graph datasets seem to doubt the positive answer to this question. The first observation is that scaling pre-training samples does not result in a one-model-fits-all increase in downstream performance (see the first row of Figure 1). Second, we observe that adding input graphs (while fixing sample size) does not improve and sometimes even deteriorates the generalization of the pre-trained model (see the second row in Figure 1). Furthermore, even if the number of input graphs (the horizontal coordinate) is fixed, the performance of the model pre-trained on different combinations of inputs varies dramatically; see the standard deviation in blue. As the first contribution, we identify the *curse of big data* phenomenon in graph pre-training: more training samples and graph datasets do not necessarily lead to better downstream performance.

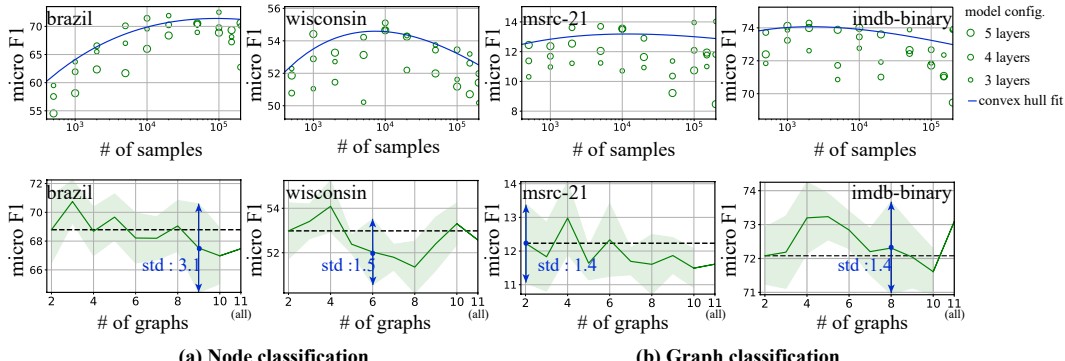

Figure 1: *Top row*: The effect of scaling up sample size (log scale) on the downstream performance based on a group of GCCs Qiu et al. (2020) under different configurations (the graphs used for pre-training are kept as all eleven pre-training data in Table 3, and the samples are taken from the backbone pre-training model according to its sampling strategy). The results for different downstream graphs (and tasks) are presented in separate figures. To better show the changing trend, we fit a curve to the best performing models (*i.e.*, the convex hull fit as Abnar et al. (2022) does). *Bottom row*: The effect of scaling up the number of graph datasets on the downstream performance based on GCC. For a fixed horizontal coordinate, we run 5 trials. For each trial, we randomly choose a combination of input graphs. The shaded area indicates the standard deviation over the 5 trials. See Appendix D for more observations on other graph pre-training models and detailed settings.

Therefore, instead of training on massive data, it is more appealing to choose wisely some samples and graphs for pre-training. However, without the knowledge of downstream tasks, the difficulty is how to design new criteria for selecting input data to the pre-training model. To fill this gap, we propose a novel *graph selector* that is able to provide the most instructive data for the model. The criteria in the graph selector include *predictive uncertainty* and *graph properties*. Predictive uncertainty is introduced to measure the level of confidence (or certainty) in the data. On the other hand, some graphs are more informative and representative than others, due to their inherent structure. To this end, some fundamental properties of graphs also help in the selection process.

Given the selected input data, we take full advantage of the predictive uncertainty as a proxy for measuring the model capability during the training phase. Instead of swallowing data as a whole, the pre-training model is encouraged to learn from the data in a progressive way. After learning a certain amount of training data, the predictive uncertainty gives feedback on what kind of data the model has least knowledge of. Then the pre-training model is able to reinforce itself on highly uncertain data in next training iterations.

Putting together, we propose a data-active graph pre-training (APT) framework, which integrates the graph selector and the pre-training model into a unified framework. The two components in the framework actively cooperate with each other. The graph selector recognizes the most instructive data for the model. Equipped with this intelligent selector, the pre-training model is well-trained and in turn provides better guidance for the graph selector.

The rest of the paper is organized as follows. In §2 we review the existing works about basic graph pre-training framework commonly used for training cross-domain graph data. Then in §3 we describe in detail the proposed data-active graph pre-training (APT) paradigm. §4 contains numerical experiments, which demonstrate the superiority of APT in different downstream tasks, especially when the test and training graphs come from different domains. Lastly, we also include the applicable scope of our pre-trained model.

## 2 BASIC GRAPH PRE-TRAINING FRAMEWORK

This section reviews the basic framework of cross-domain graph pre-training commonly used in related literature. The backbone of our graph pre-training model also follows this framework, and uses GCC Qiu et al. (2020) as an instantiation. In principle, GCC can be substituted by any encoder suitable for training cross-domain graphs.

We start with a natural question: What does cross-domain graph pre-training actually learn? Previous studies argue that the semantic meaning associated with structural patterns is transferable. For example, both in citation networks and social networks, the closed triangle structure (⬡) is interpreted as a stable relationship, while the open triangle (⬡) indicates an unstable relationship.

When data comes from other domains like molecular networks, the semantic meaning can be quite different. Nevertheless, we argue that the distinction between different structural patterns is still transferable. Taking the same example, the closed and open triangles might yield different interpretations in molecular networks (unstable vs. stable in terms of chemical property) from those in social networks (stable vs. unstable in terms of social relationship), but the distinction between these two structures remains the same because they indicate opposite (or contrastive) semantic meanings. Therefore, the cross-domain pre-training either learns representative structural patterns (when semantic meanings are present), or more importantly, obtains the capability of distinguishing these patterns. This observation in graph pre-training is not only very different from that in other areas (*e.g.*, computer vision and natural language processing), but may also explain why graph pre-training is effective, especially when some downstream information is absent.

With the hope to learn the transferable structural patterns or the ability to distinguish them, the cross-domain graph pre-training model is fed with a collection of input graphs (possibly from different domains), and the learnt model, denoted by $f_\theta$ (or simply $f$ if the parameter $\theta$ is clear from context), maps a node to a low-dimensional representation. Unaware of the specific downstream task as well as task-specific labels, one should design a self-supervised task for the pre-training model. Such self-supervised information for a node is usually hidden in its neighborhood pattern, and thus the structure of its ego network is often used as the transferable pattern. Naturally, subgraph instances sampled from the same ego network $\Gamma_i$ are considered *similar* while those sampled from different ego networks are rendered *dissimilar*. Therefore, the pre-training model attempts to capture the similarities (and dissimilarities) between subgraph instances, and such a self-supervised task is called the *subgraph instance discrimination task*. More specifically, given a subgraph instance $\zeta_i$ from an ego network $\Gamma_i$ centered at node $v_i$ as well as its representation $\boldsymbol{x}_i = f(\zeta_i)$, the model $f$ aims to encourage higher similarity between $\boldsymbol{x}_i$ and the representation of another subgraph instance $\zeta_i^+$ sampled from the same ego network. This can be done by minimizing, *e.g.*, the InfoNCE loss Oord et al. (2018):

$$\mathcal{L}_i = -\log \frac{\exp\left(\boldsymbol{x}_i^\top f(\zeta_i^+)/\tau\right)}{\exp\left(\boldsymbol{x}_i^\top f(\zeta_i^+)/\tau\right) + \sum\limits_{\zeta_i' \in \Omega_i^-} \exp\left(\boldsymbol{x}_i^\top f(\zeta_i')/\tau\right)}, \tag{1}$$

where $\Omega_i^-$ is a collection of subgraph instances sampled from different ego networks $\Gamma_j$ ($j \neq i$), and $\tau$ is a temperature hyper-parameter. Here the inner product is used as a similarity measure between two instances. One common strategy to sample these subgraph instances is via random walks on graphs, as used in GCC Qiu et al. (2020), but other sampling methods as well as loss functions are also valid.

## 3 DATA-ACTIVE GRAPH PRE-TRAINING

In this section we present the proposed APT framework for cross-domain graph pre-training, and the overall pipeline is illustrated in Figure 2. The APT framework consists of two major components, a graph selector and a graph pre-training model. The technical core is the interaction between these two components: The graph selector feeds *suitable* data for pre-training, and the graph pre-training model learns from the carefully chosen data. The feedback of the pre-training model in turn helps select the needed data tailored to the model.

The rest of this section is organized as follows. We describe the graph selector in § 3.1 and the graph pre-training model in § 3.2. The overall pre-training and fine-tuning strategy is presented in § 3.3.

### 3.1 GRAPH SELECTOR

In view of the curse of big data phenomenon, it is more appealing to carefully choose data *well suited* for graph pre-training rather than training on a massive amount of data. Conventionally, the criterion of suitable data, or the contribution of a data point to the model, is defined based on the output predictions on downstream tasks Goodfellow et al. (2016). In graph pre-training where downstream information is absent, new selection criteria or guidelines are needed to provide effective instructions for the model. Here we introduce two kinds of selection criteria, originated from different points of view, to help select suitable data for pre-training. The *predictive uncertainty* measures the model's understanding of certain data, and thus helps select the least certain data points for the current model. In addition to the measure of model's ability, some *inherent properties* of graphs can also be used to assess the level of representativeness or informativeness of a given graph.

**Predictive uncertainty.** The notion of predictive uncertainty can be explained via an illustrative example, as shown in part (a) of the graph selector component in Figure 2. Consider a query sub-

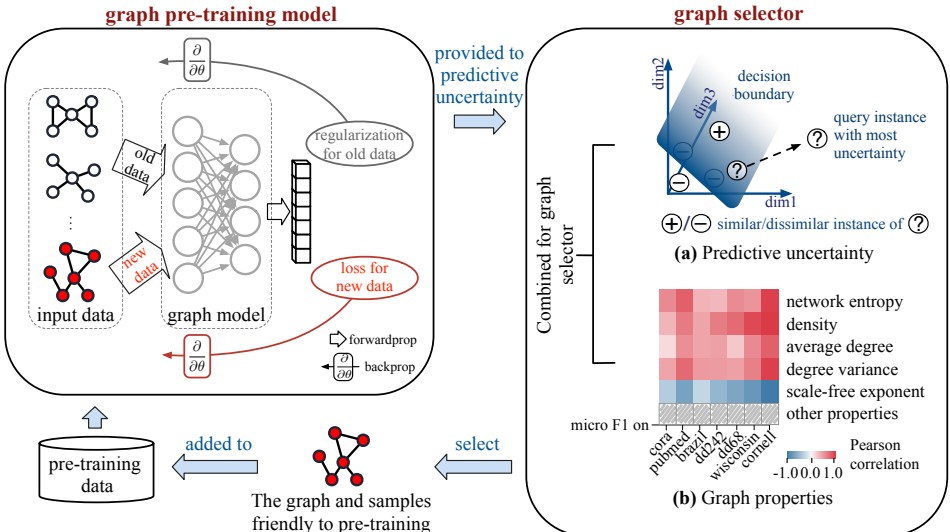

Figure 2: Overview of the proposed data-active graph pre-training paradigm. The graph selector provides the graph and samples suitable for pre-training, while the graph pre-training model learns from the incoming data in a progressive way and in turn better guides the selection process. In the graph selector component, *Part (a)* provides an illustrating example on the predictive uncertainty, and *Part (b)* plots the Pearson correlation between the properties of the input graph and the performance of the pre-trained model (using this graph) on different unseen test datasets (see Appendix E for other properties that exhibit little/correlation with performance).

graph instance $\zeta_i$ (denote by ⑦ in Figure 2) from the ego network $\Gamma_i$ in a graph $G$. If the pre-training model cannot tell its similar instance $\zeta_i^+$ (denoted by ⊕) from its dissimilar instance $\zeta_i^- \in \Omega_i^-$ (denoted by ⊖), we say that the current model is uncertain about the query instance $\zeta_i$. Therefore, the contrastive loss function in Eq. (1) comes in handy as a natural measure for the predictive uncertainty of the instance $\zeta_i$: $\phi_{\text{uncertain}}(\zeta_i) = \mathcal{L}_i$. Accordingly, the predictive uncertainty of a graph $G$ (*i.e.*, the graph-level predictive uncertainty) is defined as $\phi_{\text{uncertain}}(G) = (1/M) \sum_{i=1}^{M} \mathcal{L}_i$, where $M$ is the number of subgraph instances queried in this graph.

The proposed selection process is different from strategies used in curriculum learning Bengio et al. (2009). Predictive uncertainty encourages the model to learn more difficult (uncertain) graphs and samples in the first place, while in curriculum learning, the easiest samples are fed first. The choice of difficult-first order is intuitive in our case; see also Appendix G for empirical evidence.

**Graph properties.** As we see above, the predictive uncertainty measures the model's ability to distinguish (or identify) a given graph (or subgraph instance). However, predictive uncertainty is sometimes misleading, especially when the chosen graph (or subgraph) happens to be an outlier of the entire data collection. Hence learning solely from the most uncertain data might not improve the overall performance, or even worse, lead to overfitting. The inherent properties of the graph turn out to be equivalently important as a selection criterion for graph pre-training. Intuitively, it is preferable to choose those graphs that are *good by themselves*, those with a better structure, or those containing more information. So here we introduce five inherent properties of graphs (*i.e.*, network entropy, density, average degree, degree variance and scale-free exponent) to help select *better* data points for pre-training. All these properties exhibit a strong correlation with downstream performance, which is empirically verified and presented in part (b) of the graph selector component in Figure 2. The choice of these properties also has an intuitive explanation, and here we discuss the intuition behind the network entropy as an example.

The use of *network entropy* is inspired from the sampling methods used in most cross-domain graph pre-training models (see *e.g.*, Qiu et al. (2020); Hafidi et al. (2020)): Random walks started at a node are employed to construct a subgraph instance as the model input. Random walks can also be used to compute the amount of information contained in a graph. Especially, the amount of information contained in the move from node $v_i$ to node $v_j$ is $-\log P_{ij}$ Cover & Thomas (1999), where $P$ is the transition matrix. Thus the network entropy of a connected graph $G = (V, E)$ can be defined as the expected information of individual transitions over the random walk process Burda et al. (2009):

$$\phi_{\text{entropy}} = \langle -\log P \rangle_P = -\sum_{ij} \pi_i P_{ij} \log P_{ij}, \quad (2)$$

where $\boldsymbol{\pi}$ is the stationary distribution of a random walk and $\langle \cdot \rangle_P$ denotes the expectation of a random variable according to $P$. Network entropy (2) is in general difficult to calculate, but for a connected unweighted graph, $P_{ij} = 1/d_i$, $\boldsymbol{\pi} = (1/2|E|)\boldsymbol{d}$ (where $d_i$ is the degree of node $v_i \in V$ and $\boldsymbol{d} = (d_1, d_2, \dots)$ is the degree vector). Then the network entropy (2) reduces to

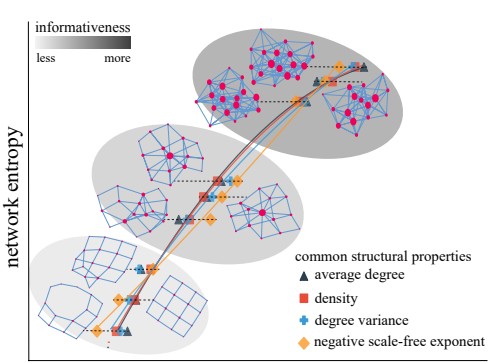

$$\phi_{\text{entropy}} = \frac{1}{2|E|} \sum_{i=1}^{N} d_i \log d_i, \quad (3)$$

where $N = |V|$ is the total number of nodes in $G$. In this case, the network entropy of a graph depends solely on its degree distribution, and is straightforward to compute.

Figure 3: The illustrative graphs (from bottom left to top right) with increasing network entropy and the other four graph properties.

Although the definition of network entropy originates from random walks on graphs, it is still useful in graph pre-training even when the sampling of subgraph instances does not depend on random walks. Here we provide another intuitive explanation of network entropy from the coding theory. Network entropy can be viewed as the entropy rate of a random walk, and it is known that the entropy rate is the expected number of bits per symbol required to describe a stochastic process Cover & Thomas (1999). Similarly, the network entropy can be interpreted as the expected number of "words" needed to describe the graph. Thus intuitively, the larger the network entropy is, the more information the graph contains.

As a final remark on network entropy, the connectivity assumption does not limit the usefulness of Eq. (3) in our case. For disconnected input graphs, we can simply compute the network entropy of the largest connected component, since for most real-world networks, the largest connected component contains most of the information Easley & Kleinberg (2010). Alternatively, we can also take some of the largest connected components from the graph and treat them separately as several connected graphs.

Furthermore, the other four graph properties, *i.e.*, density, average degree, degree variance and scale-free exponent, are closely related to the network entropy. Figure 3 presents a clear correlation between the network entropy and the other four graph properties, as well as provides some illustrative graphs. (These example graphs are generated by the configuration model proposed in Newman (2003), and Appendix E contains more results of real-world networks.) Intuitively, graphs with higher network entropy contain a larger amount of information, and so are graphs with larger density, higher average degree, higher degree variance, or a smaller scale-free exponent. The connections between all five graph properties can also be theoretically justified and the motivations of choosing these properties can be found in Appendix A. The detailed empirical justification of these properties and the pre-training performance in included in Appendix E.

**Time-adaptive selection strategy.** The proposed predictive uncertainty and the five graph properties together act as a powerful indicator of a graph's goodness. Thus the selection of graph can be formulated as the following optimization problem:

$$\text{maximize} \quad \mathcal{J}(G) = (1 - \gamma_t)\hat{\phi}_{\text{uncertain}} + \gamma_t \text{MEAN}(\hat{\phi}_{\text{entropy}}, \hat{\phi}_{\text{density}}, \hat{\phi}_{\text{avg\_deg}}, \hat{\phi}_{\text{deg\_var}}, \text{-}\hat{\phi}_\alpha), \quad (4)$$

where the optimization variable is the graph $G$ to be selected, $\hat{\phi}_{\text{uncertain}}$, $\hat{\phi}_{\text{entropy}}$, $\hat{\phi}_{\text{density}}$, $\hat{\phi}_{\text{avg\_deg}}$, $\hat{\phi}_{\text{deg\_var}}$ and $\hat{\phi}_\alpha$ are the z-score normalized value of graph-level predictive uncertainty, network entropy, density, average degree, degree variance and scale-free exponent of graph $G$ respectively, $\gamma_t \in [0, 1]$ is a parameter to trade off the predictive uncertainty and the graph properties, and $t$ is the iteration counter. Note that the pre-training model learns nothing at the beginning, so we initialize $\gamma_0 = 0$. The balance between the predictive uncertainty and the inherent graph properties ensures that the selected graph is a good supplement to the current pre-training model as well as an effective representative for the entire data distribution.

We shall also note that, at the beginning of the pre-training, the outputs of the model are not accurate enough to guide data selection, so the parameter $\gamma_t$ should be set smaller so that the graph properties play a leading role. As the training phase proceeds, the graph selector gradually pays more attention to the feedback $\phi_{\text{uncertain}}$ from the model via a larger value of $\gamma_t$. Therefore, the parameter $\gamma_t$ is called the *time-adaptive parameter*, and is set to be a random variable depending on time $t$. In this work, we take $\gamma_t$ from a Beta distribution $\gamma_t \sim \text{Beta}(1, \beta_t)$, where $\beta_t$ decreases over time (training iterations).

Finally, after a graph is selected, we can further choose subgraph instances with high predictive uncertainty for training, rather than feed the model with random subgraph samples.

**Connections and differences with hard example mining.** Hard example mining learns from the examples that contribute the most to model training, which has been widely applied in computer vision, natural language processing and recommender system Simo-Serra et al. (2014); Loshchilov & Hutter (2015); Shrivastava et al. (2016); Krishnan et al. (2020). Our usage of predictive uncertainty for choosing graphs is conceptually similar to hard example mining. However, existing approaches for hard sample mining can not be directly applied to our setting with the following two requirements. (1) The chosen instances should follow a joint distribution that reflects the topological structures of real-world graphs. This is met by our use of graph-level predictive uncertainty and graph properties. (2) The chosen set of graphs should include informative and sufficiently diverse instances. This goal is achieved by the data-active graph pre-training framework, which enables the interaction between the graph selector and pre-training model. Another line of works in active learning introduce the measure of uncertainty by querying the labels of samples that current model is least certain w.r.t classification prediction Cai et al. (2017); Zhang et al. (2021a); Yang et al. (2015); Zhu et al. (2008), which cannot be adapted in pre-training with unlabeled data.

### 3.2 GRAPH PRE-TRAINING MODEL

The graph pre-training model takes the input graphs and samples one by one and enhances itself in a progressive manner. However, such a sequential training process does not guarantee the model to *remember* all the contributions of previous input data. As shown in the orange curve in Figure 4, the previously learnt graph exhibits a larger predictive uncertainty as the training phase proceeds. The empirical result indicates that the knowledge or information contained in previous input data will be forgotten or covered by newly incoming data. This phenomenon, called *catastrophic forgetting* Kirkpatrick et al. (2017), was first noticed in continual learning and is also identified here. Intuitively, when the training data is taken in a one-by-one manner, the learnt parameters will cater to the newly incoming data compared with the old, previous data points.

One remedy for this issue is adding a proximal term to the objective. The additional proximal term (*i.e.*, the regularization) guarantees the proximity between the new parameters and the model parameters learnt from previous graph. Therefore, the final loss function for our pre-training model in APT is

$$\mathcal{L}(\theta) = \sum_i \mathcal{L}_i(\theta) + \sum_{j=k-1}^{k} \sum_m \frac{\lambda_j}{2} F_m^{(j)} \|\theta_m - \theta_m^{(j)}\|^2, \quad (5)$$

where $\mathcal{L}_i$ is given in Eq. (1), the summation in the first term is taken over the subgraph instances sampled from the new input graph, $k$ is the number of previously learnt graphs, $\theta^{(j)}$ is the model parameters learnt from the first $j$ graphs, and $\lambda_j$'s are the trade-off parameters between the knowledge learnt

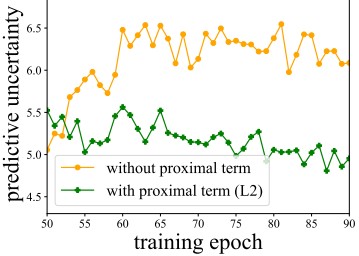

Figure 4: Predictive uncertainty of a learnt graph ("michigan") versus training epoch.

from new data and that from previous data. Typically, the trade-off parameters $\{\lambda_j\}$ form a non-decreasing sequence, *i.e.*, $\lambda_1 \leq \lambda_2 \leq \cdots \leq \lambda_k$. Inspired from the Elastic weight consolidation (EWC) algorithm Kirkpatrick et al. (2017), we take $F^{(j)}$ as the Fisher information matrix of $\theta^{(j)}$, $F_m^{(j)}$ is the diagonal element of $F^{(j)}$ and $m$ labels each parameter. When $F$ is set as an identity matrix, the second term degenerates to the L2 regularization (serves as one of our variants). Note that the proximal term in Eq. (5) is absent when the first input graph is introduced to the model.

### 3.3 TRAINING AND FINE-TUNING

Integrating the graph selector and the pre-training model forms the entire APT framework, and the overall algorithm is presented in Appendix B. After the training phase, the APT framework returns a

pre-trained GNN model, and then the pre-trained model can be applied to various downstream tasks from a wide spectrum of domains. In the so-called *freezing mode*, the pre-trained model outputted from APT is directly applied to downstream tasks, without any changes in parameters. Alternatively, the *fine-tuning mode* uses the pre-trained graph encoder as initialization, and offers the flexibility of training the graph encoder and the downstream classifier together in an end-to-end manner.

## 4 EXPERIMENTS

In the experiments, we pre-train our model on the incoming data provided by the graph selector, and then evaluate the transferability of our pre-trained model on multiple unseen graphs from different domains in the node classification and graph classification task. Lastly, we include the applicable scope of our pre-trained model. Additional experiments can be found in Appendix G, including analysis on training time, sensitivity analysis of hyper-parameters, ablation study on various combinations of graph properties.

### 4.1 EXPERIMENTAL SETUP

**Datasets.** The datasets for pre-training and testing, and their detailed statistics are listed in Appendix C. The datasets for pre-training are collected from different domains, including social networks, citation networks, and movie collaboration networks. We then evaluate the pre-trained models on 13 real-world graphs, including large-scale datasets with millions of edges from Open Graph Benchmark Hu et al. (2020a). Some of them are from the similar domain as pre-training (like citation networks), while most of them are from a totally unseen domains (like web networks, transportation networks, protein networks and others).

**Baselines.** We comprehensively evaluate our model against the following baselines for node classification and graph classification tasks, respectively. For node classification tasks, ProNE Zhang et al. (2019), DeepWalk Perozzi et al. (2014), struc2vec Ribeiro et al. (2017), DGI Velickovic et al. (2019), GAE Kipf et al. (2016), and GraphSAGE Hamilton et al. (2017) are used as baselines, and then the learned representations are fed into the logistic regression for node classification (as most of baselines did). As for graph classification tasks, we take graph2vec Narayanan et al. (2017), In-foGraph Sun et al. (2020), DGCNN Zhang et al. (2018) and GIN Xu et al. (2019) as baselines, and then feed the representations into SVM as the classifier (as most of baselines did). For both tasks, we also compare our model with (1) Random, where random vectors are generated as representations; (2) GraphCL You et al. (2020a), a GNN pre-training scheme based on contrastive learning with augmentations; (3) JOAO You et al. (2021), a GNN pre-training scheme that can automatically select data augmentations; (4) GCC Qiu et al. (2020), the state-of-the-art cross-domain graph pre-training model (the version of our model without data selection scheme, which trains on all pre-training data). GCC, GraphCL and JOAO are trained on the entire collected input data, and the suffix (rand, fine-tune) indicates that the model is trained from scratch. We also include 4 variants of our model: (1) APT-G, which removes the criteria of graph properties in the graph selector; (2) APT-P, which removes the criterion of predictive uncertainty in the graph selector; (3) APT-R, which removes the regularization w.r.t old knowledge in Eq. (5); (4) APT-L2, which degenerates the second term in Eq. (5) to the L2 regularization.

**Experimental settings.** In the training phase, we iteratively select graphs for pre-training until the predictive uncertainty of any candidate graph is below 3.5. For each selected graph, we choose those samples with predictive uncertainty higher than 3. We include $M = 500$ query subgraph instances in a graph when measuring the predictive uncertainty of this graph. The time-adaptive parameter $\gamma_t$ in Eq. (4) is drawn from $\gamma_t \sim \text{Beta}(1, \beta_t)$, where $\beta_t = 3 - 0.995^t$. We set the trade-off parameter $\lambda_j = 10$ for all $j$ for APT-L2, and $\lambda_j = 500$ for APT. The total iteration number is 100. We adopt GCC as the backbone pre-training model with their default hyper-parameters. Note that we can also use other pre-training models like GraphCL as the backbone, but we do not report them due to the non-ideal performance of GraphCL. In the fine-tuning phase, we select logistic regression or SVM as the downstream classifier and adopt the same setting as GCC. See Appendix F for more details.

### 4.2 EXPERIMENTAL RESULTS

**Node classification.** Table 1 presents the micro F1 score of different methods over 10 unseen graphs from a wide spectrum of domain for node classification task. We can observe our model beats the graph pre-training competitor by an average of 9.94% and 17.83% under freezing and fine-tuning mode respectively. This suggests that instead of pre-training on all the collected graphs (like GCC), it is better to choose a part of graphs better suited for pre-training (like our model APT). Moreover, compared with the traditional models without pre-training, the performance gain of our

Table 1: Micro F1 scores of different models in the node classification task. The column "A.R." reports the average rank of each model. Asterisk (∗) denotes the best result on each dataset, and bold numbers denote the best result among graph pre-training models in the freezing or fine-tuning setting. The notation "/" means out of memory or no convergence for more than three days.

| Method \ Dataset | brazil | dd242 | dd68 | dd687 | wisconsin | cornell | cora | pubmed | ogbarxiv | ogbproteins | A.R. |
|---|---|---|---|---|---|---|---|---|---|---|---|
| Random | 32.16(13.65) | 6.71(2.44) | 8.29(4.35) | 5.98(2.70) | 26.79(8.59) | 39.77(7.26) | 26.80(1.62) | 38.85(0.76) | 11.89(4.66) | 52.69(5.94) | 13.2 |
| ProNE | 50.24(11.56) | 10.04(2.56) | 7.73(3.11) | 3.88(1.66) | 44.67(7.49) | 47.32(12.14) | 80.76(2.92)* | 78.80(0.98)* | 65.96(0.06)* | 76.28(0.34)* | 7.5 |
| DeepWalk | 43.16(16.78) | 8.11(1.45) | 6.72(3.04) | 6.17(2.18) | 39.61(9.11) | 47.67(8.30) | 49.85(9.26) | 44.99(10.89) | 16.04 (2.98) | 64.74(0.49) | 8.7 |
| struc2vec | 25.54(11.74) | 13.71(2.66) | 10.30(3.23) | 7.98(2.74) | 45.39(7.36) | 38.39(9.18) | 36.01(2.41) | 44.45(0.78) | / | / | 10.6 |
| DGI | 56.44(7.79) | 14.35(0.44) | 13.57(0.44) | 11.04(1.93) | 49.46(5.46) | 49.85(9.26) | 30.02(0.44) | 42.39(0.84) | 12.93(7.67) | 55.98(0.33) | 6.6 |
| GAE | 57.88(10.68) | 14.09(1.52) | 13.43(0.96) | 10.25(2.63) | 45.78(4.18) | 49.26(5.24) | 30.10(0.31) | 40.14(0.68) | / | / | 8.5 |
| GraphSAGE | 67.93(9.28) | 14.33(0.37) | 13.55(0.70) | 10.39(0.78) | 47.03(1.98) | 49.20(1.46) | 35.93(1.76) | 39.94(0.02) | / | / | 7.1 |
| GraphCL (freeze) | 50.71(5.00) | 9.53(2.50) | 9.36(3.63) | 6.03(1.86) | 38.85(10.80) | 41.05(5.67) | 16.95(2.39) | 41.07(1.16) | / | / | 12.2 |
| JOAO (freeze) | 71.22(7.21) | 7.98(2.90) | 12.36(2.59) | 5.34(1.43) | 42.69(8.15) | 43.16(5.67) | 18.13(2.82) | 41.05(0.87) | / | / | 10.9 |
| GCC (freeze) | 67.47(4.09) | 15.83(0.80) | 11.95(1.13) | 9.61(0.94) | 52.57(1.69) | 46.87(1.73) | 35.47(0.51) | 46.40(0.18) | 14.56(7.60) | 59.15(0.35) | 7.0 |
| APT-G (freeze) | 68.69(3.42) | **17.21(1.13)** | 11.98(0.75) | 9.54(1.29) | 54.45(1.90) | 46.53(1.59) | 34.89(0.25) | 46.49(0.22) | 12.32(7.71) | 60.38(0.41) | 6.3 |
| APT-P (freeze) | 66.55(2.35) | 16.58(0.97) | 12.48(0.85) | 10.33(0.83) | 51.90(1.64) | 47.33(2.31) | 35.63(0.56) | 46.16(0.12) | 12.86(7.54) | 60.32(0.32) | 6.2 |
| APT-R (freeze) | 68.12(3.07) | 16.72(0.72) | 12.42(1.24) | **11.05(0.88)** | 54.48(1.77) | 46.80(1.08) | 34.93(0.36) | 46.02(0.11) | 18.79(5.87) | 62.18(0.46) | 5.0 |
| APT-L2 (freeze) | 69.82(2.32) | 16.79(0.88) | **12.68(0.81)** | 10.34(1.12) | **55.11(1.74)** | **48.76(2.20)** | 34.27(0.43) | 46.21(0.15) | 19.64(6.46) | 60.23(0.37) | 4.4 |
| APT (freeze) | **73.39(2.55)** | 16.57(0.94) | 12.08(0.89) | 10.35(1.24) | 53.38(1.19) | 47.37(1.29) | **36.69(0.49)** | **46.88(0.21)** | **22.04(0.29)** | **62.29(0.55)** | 3.8 |
| GraphCL (rand, fine-tune) | 64.43(14.95) | 15.04(0.85) | 14.69(2.48) | 10.99(0.58) | 63.85(2.18) | 44.21(10.58) | 30.45(0.37) | 40.73(0.66) | / | / | 8.3 |
| JOAO (rand, fine-tune) | 72.14(6.74) | 10.93(2.85) | 8.08(2.15) | 7.40(3.48) | 45.38(13.30) | 45.26(10.31) | 29.93(2.84) | 42.01(0.68) | / | / | 9.6 |
| GCC (rand, fine-tune) | 58.51(3.07) | 15.98(1.05) | 13.16(1.06) | 9.74(0.95) | 53.85(2.58) | 50.95(2.26) | 43.70(0.52) | 49.72(0.17) | 18.61(1.88) | 59.12(0.35) | 7.6 |
| GraphCL (fine-tune) | 73.57(10.33) | 15.35(0.99) | 13.51(2.57) | 10.66(1.04) | 63.85(4.42) | 51.05(2.41) | 30.81(0.36) | 42.91(0.91) | / | / | 7.5 |
| JOAO (fine-tune) | 75.00(5.76) | 10.54(3.07) | 7.56(1.94) | 8.77(2.39) | 50.0(12.28) | 42.11(10.26) | 29.34(3.04) | 42.21(0.88) | / | / | 9.5 |
| GCC (fine-tune) | 74.46(3.05) | 19.32(0.80) | 13.87(1.13) | 10.37(1.06) | 59.47(1.49) | 48.32(2.42) | 43.34(0.38) | 50.87(0.19) | 18.62(1.92) | 60.08(0.56) | 6.4 |
| APT-G (fine-tune) | 77.60(1.48) | 25.45(0.60)* | 17.78(1.14) | 11.27(0.76) | 66.09(2.28) | 53.02(1.51) | 45.63(0.66) | 50.81(0.18) | 27.33(4.80) | 60.02(0.32) | 3.8 |
| APT-P (fine-tune) | 78.99(2.44) | 25.19(0.87) | 16.40(1.22) | 11.69(1.19) | 64.24(1.90) | 50.05(1.39) | 45.53(0.30) | 50.66(0.18) | 27.20(4.80) | 59.86(0.32) | 4.8 |
| APT-R (fine-tune) | 79.14(1.97) | 24.96(0.57) | 17.43(1.05) | 11.29(1.04) | 66.28(1.94) | 53.56(2.28)* | 46.02(0.83) | 51.00(0.21) | 18.41(1.84) | 60.10(0.38) | 3.4 |
| APT-L2 (fine-tune) | 78.75(1.63) | 24.62(0.90) | 17.83(1.35)* | 12.26(0.78)* | 67.04(1.50)* | 52.94(1.95) | 47.48(0.46) | 51.25(0.21) | 27.40(4.97) | 60.85(0.46) | 2.6 |
| APT (fine-tune) | **79.67(2.30)*** | **28.62(0.55)*** | **20.30(1.13)*** | **12.80(1.54)*** | **67.08(1.75)*** | 52.15(2.25) | **47.51(0.62)** | **51.30(0.16)** | **27.40(4.87)** | **61.64(0.35)** | 1.3 |

model is attributed to the transferable knowledge learned by pre-training strategies. We also find that some proximity-based models like ProNE enforce neighboring nodes share similar representations, thus they perform well on graphs with strong homophily rather than weak homophily.

**Graph classification.** The micro F1 score on unseen test data in the graph classification task is summarized in Table 2. Especially, our model is 7.2% and 1.3% on average better than the graph pre-training backbone model under freezing and fine-tuning mode, respectively. Interestingly, we find that the variants of APT perform well under graph classification, indicating that we can apply a version with simpler architecture in practice yet achieve good results.

**Analysis of the selected graphs.** The data sequentially selected via our graph selector are uillinois, soc-sign0811, msu, michigan, wiki-vote, soc-sign0902 and dblp. To further analyze why these graphs are chosen, we present their detailed structural properties in Table 4 in the Appendix C. We first observe that uillinois, michigan and msu have the largest value of

Table 2: Micro F1 of different models in the graph classification.

| Method \ Dataset | imdb-binary | dd | msrc-21 | A.R. |
|---|---|---|---|---|
| Random | 49.30(4.82) | 52.72(4.34) | 4.49(2.14) | 11 |
| graph2vec | 56.20(5.33) | 59.16(3.47) | 8.22(3.67) | 7.7 |
| InfoGraph | 66.58(0.63) | 58.66(0.23) | 6.01(0.59) | 7.7 |
| GraphCL (freeze) | 55.10(3.18) | 57.82(4.71) | 5.44(2.77) | 9.3 |
| JOAO (freeze) | 63.90(3.48) | 55.97(3.61) | 5.09(2.65) | 9.3 |
| GCC (freeze) | 73.09(0.55) | 75.16(0.53) | 11.61(1.33) | 5.3 |
| APT-G (freeze) | 73.10(0.39) | 75.24(0.42) | 12.81(0.74) | 4.3 |
| APT-P (freeze) | 72.83(0.81) | **76.38(0.32)** | 13.30(0.57) | 3.0 |
| APT-R (freeze) | **73.98(0.21)** | 75.32(0.34) | 12.90(0.57) | 3.0 |
| APT-L2 (freeze) | 73.54(0.40) | 75.81(0.30) | 13.16(0.77) | 2.3 |
| APT (freeze) | 73.00(0.50) | 75.83(0.31) | **13.81(1.06)** | 3.0 |
| DGCNN | 71.00(4.69) | 58.63(4.46) | 6.01(0.59) | 11.3 |
| GIN | 72.00(2.41) | 77.61(1.47)* | 10.54(5.08) | 6.0 |
| GraphCL (rand, fine-tune) | 63.60(3.61) | 58.15(4.60) | 8.25(2.94) | 12.7 |
| JOAO (rand, fine-tune) | 67.70(3.35) | 62.10(4.31) | 11.40(3.06) | 10.0 |
| GCC (rand, fine-tune) | 75.80(1.37) | 74.26(0.59) | 17.18(1.43) | 7.3 |
| GraphCL (fine-tune) | 66.90(4.39) | 65.55(5.14) | 8.77(2.60) | 10.7 |
| JOAO (fine-tune) | 68.50(3.61) | 62.61(4.99) | 10.18(1.72) | 10.0 |
| GCC (fine-tune) | 76.19(0.90) | 75.32(1.77) | 24.90(1.65) | 4.7 |
| APT-G (fine-tune) | 76.29(0.89) | 75.46(0.77) | 21.94(0.73) | 4.7 |
| APT-P (fine-tune) | **76.70(1.01)*** | 75.34(0.88) | 24.32(1.22) | 3.7 |
| APT-R (fine-tune) | 76.60(1.02) | 75.64(0.70) | 24.09(2.12) | 3.3 |
| APT-L2 (fine-tune) | 75.93(0.84) | 75.58(1.06) | 25.58(1.57)* | 3.7 |
| APT (fine-tune) | 76.27(1.20) | **75.69(1.42)** | 24.41(1.82) | 3.0 |

$\text{MEAN}(\hat{\phi}_{\text{entropy}}, \hat{\phi}_{\text{density}}, \hat{\phi}_{\text{avg\_deg}}, \hat{\phi}_{\text{deg\_var}}, -\hat{\phi}_{\alpha})$, while dblp has the smallest. This shows that both criteria, the graph properties and the predictive uncertainty, play an important role in data selection. Moreover, it is also interesting to see that wiki-vote is the smallest graph among all the pre-training graphs, but it still contributes to the performance. This observation again verifies the curse of big data phenomenon in graph pre-training.

## 4.3 DISCUSSION: SCOPE OF APPLICATION

The transferability of the pre-trained model comes from the learnt representative structural patterns and the ability to distinguish these patterns (as discussed in §2). Therefore, our pre-training model is more suitable for the datasets where the target (*e.g.*, labels) is correlated with subgraph patterns or structural properties (*e.g.*, motifs, triangles, betweenness, stars). For example, for node classification on heterophilous graphs (*e.g.*, winconsin, cornell), our model performs very well because in these graphs, nodes with the same label are not directly connected, but share similar structural properties and behavior (or role, position). On the contrary, graphs with strong homophily (like cora, pubmed, ogbarxiv and ogbproteins) may not benefit too much from our models. Similar observation can

also be made on graph classification: our model could also benefit the graphs whose label has a strong relationship with their structure, like molecular, chemical, and protein networks (e.g., dd in our experiments) Vishwanathan et al. (2010); Gardiner et al. (2000).

## 5 RELATED WORKS

**Pre-training in CV and NLP.** Initially, the CV community benefits from the models like Vision Transformers Liu et al. (2021), MLP-mixers Tolstikhin et al. (2021) and ResNets He et al. (2016), which are supervised pre-trained on large-scale image data. To take full advantage of massive unlabeled data, NLP community adapts self-supervised learning models like Transformer-based encoder Vaswani et al. (2017); Radford & Narasimhan (2018); Devlin et al. (2019) for language pre-training. When pre-training in CV and NLP, researchers find that scaling up the pre-training data size would results in a better or saturating performance in downstream Tan & Le (2019); Kaplan et al. (2020); El-Nouby et al. (2021); Abnar et al. (2022); Raffel et al. (2020). However, this is not true in graph pre-training. In this paper we argue that adding input graphs or pre-training samples does not necessarily improve, but sometimes even deteriorates the downstream performance.

In view of the above phenomenon in CV and NLP pre-training, data selection is not an active research direction. The only related research we notice focus on domain-specific pre-training models, which select pre-training data that is most similar to the downstream domain Cui et al. (2018); Beltagy et al. (2019); Dai et al. (2019; 2020); Yan et al. (2020); Lee et al. (2020); Chakraborty et al. (2022). The assumption on the knowledge of downstream domain is different from the across-domain graph pre-training in our paper, and thus data selection in CV/NLP pre-training is not that relevant to the current work.

**Graph pre-training.** Taking inspiration from the pre-training in CV and NLP, recent efforts have shed the light on pre-training GNNs. Initially, some unsupervised graph representation learning can be used for graph pre-training Tang et al. (2015); He et al. (2016); Grover & Leskovec (2016); Narayanan et al. (2017); Ribeiro et al. (2017); Donnat et al. (2018); Zhang et al. (2019); Hamilton et al. (2017). They are designed based on neighborhood similarity assumption, thus cannot generalize to unseen nodes and graphs. Later, a line of graph self-supervised learning can be also treated as graph pre-training, which are categorized into two folds: graph generative models and contrastive models. Graph generative models capture the universal graph patterns by recovering certain parts of input graph (*e.g.*, masked structure or attributes) Kipf et al. (2016); Wang et al. (2017); Hu et al. (2020c); Cui et al. (2020); Hou et al. (2022). These works rely on specific domain knowledge, for example the node/edge/attribute type should be the same, which makes them difficult to transfer across different types of graphs.*f* On the other hand, graph contrastive models maximize the agreement between positive pairs and minimize that between negative pairs Velickovic et al. (2019); Hu et al. (2020b); You et al. (2020a); Zhu et al. (2020); Hassani & Khasahmadi (2020); Sun et al. (2020); Li et al. (2021); Lu et al. (2021); Sun et al. (2021); Zhu et al. (2021b); Xu et al. (2021); Zhu et al. (2021a); Lee et al. (2022); Zeng & Xie (2021); Zhang et al. (2021b); Han et al. (2022). One of the technical cores is to design appropriate data augmentation like attribute masking, edge perturbation, node dropping, diffusion, *etc.*, which either performed on node attributes or the whole graph structure. So they only achieve transferability in graphs from similar (or the same) domains, or the downstream task is restricted to graph classification. With the purpose of learning transferable patterns across different domains, some works take subgraph sampling as augmentation, such that the transferable (sampled) subgraph patterns can be captured during pre-training Qiu et al. (2020); You et al. (2020a; 2021). However, these existing works only focus on how to design the pre-training model, rather than how to select data for pre-training. Our paper first points out the necessity of selecting data, and fills the gap of the data selection strategy in graph pre-training.

## 6 CONCLUSION

In this paper, we observe that big data is not a necessity for pre-training GNNs. This motivates us to wisely choose some suitable graphs and samples for pre-training rather than training on a massive amount of data. Without any knowledge of the downstream tasks, we propose a novel graph selector to provide the most instructive data for the model. The pre-training model is then encouraged to learn from the data in a progressive way and reinforce itself on newly selected data. We integrate the graph selector and the graph pre-training model in a unified framework, and form a data-active graph pre-training (APT) paradigm. The two components in APT are able to mutually boost the capability of each other. Extensive experimental results show that the proposed APT framework can enhance model capability with a fewer number of input data.

## 7 REPRODUCIBILITY STATEMENT

We provide an open-source implementation of our model APT at `https://github.com/anonymous-APT-ai/Anonymous-APT-code`. Hyperparameters necessary for reproducing the experiments can be found in §4.1, Appendix D and Appendix F. Users can run APT on their own datasets.

## 8 ETHICS STATEMENT

This paper empirically identifies the curse of big data phenomenon in graph pre-training, and develops a novel data-active graph pre-training framework. All the experiments are conducted on publicly available datasets for reproducibility purposes. Overall, this work inherits some of the risks of the existing works implementing these pre-existing datasets, and the risks are not amplified by the work. Therefore, this present paper likely does not introduce any new ethical or future social concerns.

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

# A  THEORETICAL CONNECTION BETWEEN NETWORK ENTROPY AND TYPICAL GRAPH PROPERTIES

Many interesting graph structural properties from basic graph theory give rise to a graph with high network entropy Lynn et al. (2020). We here theoretically show some connections between the proposed network entropy and typical structural properties.

To make theoretical analysis, we consider connected, unweighted and undirected graph, whose network entropy depends solely on its degree distribution (see Eq. (3)). Considering a random graph $G$ with a fixed node set, we suppose that the degree of any node $v_i$ independently follows distribution $p$, which is a common setting in random graph theory Gómez-Gardenes & Latora (2008). Then the expected network entropy of $G$ is

$$\langle \mathcal{H}(G) \rangle = \frac{1}{2|E|} \sum_i \langle \boldsymbol{d}_i \log \boldsymbol{d}_i \rangle = \frac{\langle \boldsymbol{d} \log \boldsymbol{d} \rangle}{\langle \boldsymbol{d} \rangle}. \tag{6}$$

where every $\boldsymbol{d}_i$ (and $\boldsymbol{d}$) is an independent random variable follows the distribution $p$.

Now we are ready to discuss the connection between network entropy $\langle \mathcal{H}(G) \rangle$ and some typical graph properties (*i.e.*, average degree $\langle \boldsymbol{d} \rangle$, degree variance $\mathrm{Var}(\boldsymbol{d})$ and the scale-free exponent $\alpha$).

**Average degree.**  Given that the function $x \log x$ is convex in $x$, we have

$$\langle \mathcal{H}(G) \rangle \geq \frac{\langle \boldsymbol{d} \rangle \log \langle \boldsymbol{d} \rangle}{\langle \boldsymbol{d} \rangle} = \log \langle \boldsymbol{d} \rangle. \tag{7}$$

It is clear that average degree is the lower bound of network entropy. Based on our discussion on §3.1, we conclude that when used for pre-training, an input graph with higher average degree would in general result in better performance of the pre-trained model.

**Degree variance.**  The Taylor expansion of $\langle \boldsymbol{d} \log \boldsymbol{d} \rangle$ in Eq. (6) at $\langle \boldsymbol{d} \rangle$ gives

$$\langle \mathcal{H}(G) \rangle = \log \langle \boldsymbol{d} \rangle + \frac{\mathrm{Var}(\boldsymbol{d})}{2 \langle \boldsymbol{d} \rangle^2} + o \left( \frac{1}{\langle \boldsymbol{d} \rangle^2} \right).$$

where $\mathrm{Var}(\boldsymbol{d})$ is the variance of $\boldsymbol{d}$. We find that $\log \langle \boldsymbol{d} \rangle$ is exactly the zeroth-order term in the expansion. When average degree is fixed, the network entropy and the degree variance $\mathrm{Var}(\boldsymbol{d})$ are positively correlated. This in turn implies a positive correlation between degree variance and the test performance of the model.

**Scale-free exponent.**  Most real-world networks exhibit an interesting *scale-free* property (*i.e.*, only a few nodes have high degrees), and thus the degree distribution often follows a power-law distribution. That is, we can just write the degree distribution as $p(x) \sim x^{-\alpha}$, where $\alpha$ is called the *scale-free exponent*. For a real-world network, the scale-free exponent $\alpha$ is usually larger than 2 Clauset et al. (2009). Suppose the degrees of a random graph $G$ with $N$ nodes follows a power-law distribution $p(x) = Cx^{-\alpha}$ where $C$ is a normalization constant. When $\alpha > 2$, we could approximately have Gómez-Gardenes & Latora (2008)

$$\langle \mathcal{H}(G) \rangle = \frac{1}{\alpha - 2}, \quad \text{if } N \to \infty.$$

Clearly, a smaller scale-free exponent $\alpha$ results in a higher network entropy.

**Remark 1 (Connection between network entropy and typical structural properties)** *A  graph with high network entropy arises from graphs with typical structural characteristics like large average degree, large degree variance, and scale-free networks with low scale-free exponent.*

Besides the above theoretical analysis. The motivation of choosing density, average degree, degree variance and scale-free exponent is similar to that of network entropy. Intuitively, graphs with larger *average degree* and higher *density* have more interactions among the nodes, thus providing more topological information to graph pre-training. Also, the larger the diversity of node degrees, the more diverse the subgraph samples. The diversity of node degrees can be measured by *degree variance* and *scale-free exponent*. (A smaller scale-free exponent indicates the length of the tail of degree distribution is relatively longer, *i.e.*, the degree distribution spreads out wider. )

## B   ALGORITHM

The overall algorithm for APT is given in Algorithm 1. Given a collection of graphs $\mathcal{G} = \{G_1, \ldots, G_N\}$ from various domains, APT aims to pre-train a better generalist GNN (*i.e.*, pre-training model) on wisely chosen graphs and samples. Our APT pipeline involves the following three steps. (i) At the beginning, the graph selector chooses a graph for pre-training according to the graph properties (line 1). (ii) Given the chosen graph, the graph selector chooses the subgraph samples whose predictive uncertainty is higher than $T_s$ in this graph (line 3). (iii) The selected samples are then fed into the model for pre-training until the predictive uncertainty of the chosen graph is below $T_g$ or the number of training iterations on this chosen graph reaches $F$ (line 4-5). (iv) The model's feedback in turn helps select the most needed graph based on predictive uncertainty and graph properties until the predictive uncertainty of any candidate graph is low enough (line 6-7). The last three steps are repeated until the iteration number reaches a pre-set maximum value T (which can be considered as the total iteration number required to train on all selected graphs).

---

**Algorithm 1** Overall algorithm for APT.

---

**Input:** A collection of graphs $\mathcal{G} = \{G_1, \ldots, G_N\}$, maximal period $F$ of training one graph, trade-off parameter $\gamma_t = 0$, hyperparameter $\{\beta_t\}$, the learning rate $\mu$, the predictive uncertainty threshold $T_g$ of moving to a new graph, the predictive uncertainty threshold $T_s$ of choosing training samples, and the maximum iteration number $T$.
**Output:** Model parameter $\theta$ of the pre-trained graph model.

1: Choose a graph $G^*$ from $\mathcal{G}$ via the graph selector, and $\mathcal{G} \leftarrow \mathcal{G} \backslash \{G^*\}$.
2: **while** The iteration number reaches $T$ **do**
3:    Sample instances with predictive uncertainty higher than $T_s$ from $G^*$ via the graph selector.
4:    Update model parameters $\theta \leftarrow \theta - \mu \nabla_\theta \mathcal{L}(\theta)$.
5:    **if** $\phi_{\text{uncertain}}(G^*) < T_g$ *or* the model has been trained on $G^*$ by $F$ iterations **then**
6:       Update the trade-off parameter $\gamma_t \sim \text{Beta}(1, \beta_t)$.
7:       Choose a graph $G^*$ from $\mathcal{G}$, and $\mathcal{G} \leftarrow \mathcal{G} \backslash \{G^*\}$.
8:    **end if**
9: **end while**

---

The time complexity of our model mainly consists of five components: data augmentation, GNN encoder propagation, contrastive loss, sample selection and graph selection. Suppose the maximal number of nodes of subgraph instances is $|V|$, the batch size is $B$, and $D$ is the representation dimension. (1) As for the data augmentation, the time complexity of random walk with restart is at least $O(B|V|^3)$ Xia et al. (2019). (2) The time complexity of GNN encoder propagation depends on the architectures of the backbone GNN. We denote it as $X$ here. (3) The time complexity of the contrastive loss is $O(B^2D)$ Li et al. (2022). (4) Sample selection is conducted by choosing the samples with high contrastive loss (the loss is computed before), which costs $O(B)$. (5) Graph selection costs $O(|\mathcal{G}|M^2D)$ (where $M$ the number of samples needed to compute the predictive uncertainty of a graph, and $|\mathcal{G}|$ is the number of graphs that have not been selected). This step is executed only in a few epochs (around 6% in our current model), so we ignore its time overhead in graph selection. Therefore, the overall time complexity of APT in each batch is $O(B|V|^3 + X + B^2D + B)$.

## C   DATASET DETAILS

The graph datasets for pre-training and testing in this paper are collected from a wide spectrum of domains (see Table 3 for an overview). The consideration of the graphs for pre-training and test is as follows. When selecting pre-training data, we hope that the graph size is at least hundreds of thousands to contain enough information for pre-training. When selecting test data, we hope that: (1) some test data is in the same domain as the pre-training data, and some is cross-domain, so as to comprehensively evaluate our model's in and across-domain transferability. Accordingly, the in-domain test data is selected from the type of movie and citations, and the others test data are across-domain; (2) the size of test graphs can scale from hundreds to millions.

Regarding the pre-training datasets, arxiv, dblp and patents-main are citation networks collected from Bonchi et al. (2012), Yang & Leskovec (2012) and Hall et al. (2001), respectively. Imdb is the collection of movie from Rossi & Ahmed (2015). As for the social networks, soc-sign0902 and soc-sign0811 are collected from Leskovec et al. (2009), wiki-vote is from Leskovec et al. (2010), academia is from Fire et al. (2011), and michigan, msu and uillions are from Traud et al. (2012). Regarding the test datasets, we collect the protein network dd and ogbproteins from Dobson & Doig (2003) and Hu et al. (2020a). The image network msrc-21 is from Neumann et al. (2016). The movie network imdb-binary is from Yanardag & Vishwanathan (2015). The citation networks, cora, pubmed and ogbarxiv, are from McCallum et al. (2000), Namata et al. (2012) and Hu et al. (2020a). The web networks cornell and wisconsin are collected from Pei et al. (2019). The transportation network brazil is form Ribeiro et al. (2017), and dd242, dd68 and dd687 are from Rossi & Ahmed (2015).

The detailed graph properties of the pre-training data and test data are presented in Table 4 and Table 5, respectively.

Table 3: Datasets for pre-training and testing, where $*$ denotes the average statistic of multiple graphs under graph classification setting. $|V|$ and $|E|$ denote the number of nodes and the number of edges in a graph, respectively.

| | Type | Name | $|V|$ | $|E|$ | Description |
|---|---|---|---|---|---|
| **pre-training data** | citations | arxiv | 86,376 | 517,563 | citations between papers on the arxiv |
| | | dblp | 93,156 | 178,145 | same as above (dblp) |
| | | patents-main | 240,547 | 560,943 | citations between US patents |
| | social | soc-sign0902 | 81,867 | 497,672 | friend/foe links between the users of Slashdot in Feb. 2009 |
| | | soc-sign0811 | 77,350 | 468,554 | same as above (Nov. 2008) |
| | | wiki-vote | 7,115 | 100,762 | voting relationships between wikipedia users |
| | | academia | 137,969 | 369,692 | friendships between academics on Academia.edu |
| | | michigan | 30,147 | 1,176,516 | friendships between Facebook users in University of Michigan |
| | | msu | 32,375 | 1,118,774 | same as above (Michigan State University) |
| | | uillinois | 30,809 | 1,264,428 | same as above (University of Uillinois) |
| | movie | imdb | 896,305 | 3,782,447 | relationships between actors and movies |
| **test data** | protein | dd | 284.32$*$ | 715.66$*$ | molecular interactions between amino acids |
| | | ogbproteins | 132,534 | 39,561,252 | biologically associations between proteins |
| | image | msrc-21 | 77.52$*$ | 198.32$*$ | adjacency between superpixels of the image segmentations |
| | movie | imdb-binary | 19.77$*$ | 96.53$*$ | collaboration relationships between actors/actresses |
| | citations | cora | 2,708 | 5,278 | citations between Machine Learning papers |
| | | pubmed | 19,717 | 88,648 | citations between scientific papers |
| | | ogbarxiv | 169,343 | 1,166,243 | citation network between computer science arXiv papers |
| | web | cornell | 183 | 280 | hyperlinks between webpages collected from Cornell University |
| | | wisconsin | 251 | 466 | same as above (Wisconsin University) |
| | transportation | brazil | 131 | 2,077 | commercial flights between airports in Brazil |
| | others | dd242 | 1,284 | 3,303 | this network dataset is in the category of labeled networks |
| | | dd68 | 775 | 2,093 | same as above |
| | | dd687 | 725 | 2,600 | same as above |

Table 4: Detailed structural properties of pre-training datasets, where avg properties equals to MEAN($\hat{\phi}_{\text{entropy}}, \hat{\phi}_{\text{density}}, \hat{\phi}_{\text{avg\_deg}}, \hat{\phi}_{\text{deg\_var}}, -\hat{\phi}_{\alpha}$) in Eq. (4), and $nei_2$ denotes the average number and standard deviation of $2-$hop neighbors, $|V|$ and $|E|$ denote the number of nodes and the number of edges in a graph, respectively.

| Dataset | $|V|$ | $|E|$ | avg properties | avg degree | degree var | density | entropy | $\alpha$ | $nei_2$ (avg, std) | avg clustering coef |
|---|---|---|---|---|---|---|---|---|---|---|
| soc-sign0902 | 81867 | 497672 | **-0.32** | 13.16 | 1643.20 | 1.49e-04 | 3.91 | 1.51 | 1192.33, 2305.99 | 0.06 |
| soc-sign0811 | 77350 | 468554 | **-0.32** | 13.12 | 1631.77 | 1.57e-04 | 3.93 | 1.52 | 1226.93, 2312.03 | 0.05 |
| imdb | 896305 | 3782447 | **-0.66** | 9.44 | 298.27 | 9.42e-06 | 3.07 | 1.53 | 316.16, 614.03 | 5e-05 |
| patent | 240547 | 560943 | **-0.96** | 5.66 | 34.95 | 1.94e-05 | 2.04 | 1.57 | 117.23, 172.40 | 0.08 |
| academia | 137969 | 369692 | **-0.89** | 6.36 | 102.14 | 3.88e-05 | 2.38 | 1.57 | 101.40, 225.96 | 0.14 |
| wiki-Vote | 7115 | 103689 | **0.74** | 29.32 | 3314.79 | 4.10e-03 | 4.46 | 1.40 | 972.03, 1045.13 | 0.14 |
| dblp | 93156 | 178145 | **-1.12** | 4.82 | 58.05 | 4.11e-05 | 2.16 | 1.72 | 58.85, 90.46 | 0.27 |
| arxiv | 86376 | 517563 | **-0.43** | 12.98 | 382.12 | 1.39e-04 | 3.22 | 1.41 | 145.21, 309.12 | 0.68 |
| michigan | 30147 | 1176516 | **1.40** | 79.05 | 6369.17 | 2.59e-03 | 4.78 | 1.23 | 4683.90, 3655.25 | 0.21 |
| msu | 32375 | 1118774 | **1.13** | 70.11 | 5087.53 | 2.13e-03 | 4.62 | 1.23 | 4567.63, 3465.16 | 0.21 |
| uillinois | 30809 | 1264428 | **1.44** | 83.08 | 6306.02 | 2.66e-03 | 4.78 | 1.22 | 5267.10, 3831.37 | 0.21 |

Table 5: Detailed structural properties of test datasets, where $nei_2$ denotes the average number and standard deviation of $2-$hop neighbors, and the numbers with $*$ denote the average statistics of multiple graphs under graph classification setting. $|V|$, $|E|$ and $|G|$ denote he number of nodes in a graph, the number of edges in a graph and the number of graphs in graph classification datasets, respectively.

| Dataset \ Properties | $|V|$ | $|E|$ | $|G|$ | avg degree | degree var | density | entropy | $nei_2$ (avg, std) | avg clustering coef | # of classes |
|---|---|---|---|---|---|---|---|---|---|---|
| imdb-binary | 19.77* | 193.06* | 1000 | 9.89* | 116.01* | 1.04* | 1.07* | 24.89*,15.91* | 0.95* | 2 |
| msrc-21 | 77.52* | 198.32* | 563 | 6.10* | 30.26* | 6.81e-02* | 1.71* | 17.00*, 5.81* | 0.51* | 20 |
| dd | 284.32* | 715.66* | 1178 | 6.00* | 27.60* | 2.78e-02* | 1.65* | 14.30*,5.68* | 0.48* | 2 |
| cora | 2708 | 5278 | / | 4.90 | 42.53 | 1.44e-03 | 1.71 | 34.98,47.70 | 0.24 | 7 |
| pubmed | 19717 | 44327 | / | 5.50 | 75.44 | 2.28e-04 | 2.23 | 57.10,82.72 | 0.06 | 3 |
| brazil | 131 | 1074 | / | 16.85 | 539.18 | 1.26e-01 | 3.14 | 92.27,28.50 | 0.66 | 4 |
| dd242 | 1284 | 3303 | / | 6.14 | 28.80 | 4.01e-03 | 1.68 | 14.57,4.30 | 0.47 | 20 |
| dd68 | 775 | 2093 | / | 6.40 | 33.42 | 6.98e-03 | 1.76 | 17.40,8.93 | 0.44 | 20 |
| dd687 | 725 | 2600 | / | 8.17 | 55.78 | 9.91e-03 | 2.01 | 25.45,9.96 | 0.48 | 20 |
| wiscosin | 251 | 466 | / | 4.65 | 76.26 | 1.49e-02 | 1.84 | 68.04,58.22 | 0.23 | 5 |
| cornell | 183 | 280 | / | 4.04 | 58.48 | 1.68e-02 | 1.74 | 54.09,44.30 | 0.18 | 5 |
| ogbarxiv | 16343 | 1157799 | / | 14.67 | 4898.17 | 8.07e-05 | 3.63 | 3483.08,6711.40 | 0.23 | 40 |
| ogbproteins | 132534 | 39561252 | / | 598.00 | 742637.58 | 4.50e-03 | 6.84 | 32265.17,19401.46 | 0.28 | 2 |

# D    ADDITIONAL OBSERVATIONS OF *Curse of Big Data* PHENOMENON

This section provides more comprehensive observations to support the *curse of big data* phenomenon in cross-domain graph pre-training, *i.e.*, more training samples and graph datasets do not necessarily lead to better downstream performance.

We investigate 3630 experiments with GCC Qiu et al. (2020) and GraphCL Hafidi et al. (2020) model with different model configurations (*i.e.*, the number of GNN layers is set to be 3, 4 and 5 respectively), when pre-trained on all training graphs listed in Table 3 and evaluated on different test graphs (annotated in the upper left corner of each figure) under freezing setting. Note that GCC and GraphCL are the only two pre-training models that can be adopted for the cross-domain setting. For each experiment, we calculate the mean and standard deviation over 10 evaluation results of the downstream task with random training/testing splits.

The observations of GCC and GraphCL model can be found in Figure 5 and Figure 6 respectively. The downstream results of different test data are presented in separate rows. The figures in left three columns present the effect of scaling up the number of graphs on the downstream performance under different model configurations (*i.e.*, the number of GNN layers) respectively. We first pre-train the model with only two input graphs, and the result is plotted in a dotted line. The largest standard deviation among the results w.r.t different graph last is also marked by the blue arrow. The figures in the last column illustrate the effect of scaling up sample size (log scale) on the performance.

Table 6: The value of parameters for fitting the curve according to the function $f(x) = a_1 \ln x / x^{a_2} + a_3$ ($a_1, a_2, a_3 > 0$), based on the points in the last column in Figure 5 and Figure 6.

| Dataset \ Parameter | GCC | | | GraphCL | | |
|---|---|---|---|---|---|---|
| | $a_1$ | $a_2$ | $a_3$ | $a_1$ | $a_2$ | $a_3$ |
| cora | 0.45 | 0 | 33.43 | 1038.19 | 1.24 | 17.15 |
| pubmed | 4.74 | 0.11 | 39.63 | 3.50 | 0.12 | 36.83 |
| brazil | 39.10 | 0.09 | 0 | 29.59 | 0.19 | 41.83 |
| dd242 | 6.60 | 0.08 | 3.82 | 7.84 | 0.12 | 0 |
| dd68 | 6.44 | 0.11 | 3.83 | 2.08e+17 | 6.55 | 10.37 |
| dd687 | 968.01 | 1.37 | 10.31 | 5.21 | 0.12 | 1.35 |
| wisconsin | 26.78 | 0.11 | 16.59 | 11.42 | 0.21 | 38.24 |
| cornell | 15.46 | 0.41 | 45.45 | 7.84 | 4.83 | 51.79 |
| imdb-binary | 7.43 | 0.13 | 64.97 | 1.06 | 0 | 51.69 |
| dd | 0.81 | 0.31 | 75.53 | 13.76 | 0.10 | 36.33 |
| msrc | 5.56 | 0.11 | 4.85 | 4.71 | 0.13 | 0 |

**The explanation of convex hull fit.** In order to better show the changing trend, the blue curve in the last column in Figure 5 and Figure 6 is fitted to the convex hull of the points. The convex hull is proposed to capture the performance of a randomized classifier made by choosing pre-training models with different probabilities Abnar et al. (2022).

We first introduce the concept of *randomized classifier*. Given two classifiers with training sample size and downstream performance $c_1 = (c_1^{\text{sz}}, c_1^{\text{ds}})$ and $c_2 = (c_2^{\text{sz}}, c_2^{\text{ds}})$, a *randomized classifier* can be made to choose the first classifier with probability $p$ and the second classifier with probability $1 - p$. Then the output of the randomized classifier is $pc_1 + (1 - p)c_2$, which is the convex combination of $c_1$ and $c_2$. All the points on this convex combination can be obtained by choosing different $p$. Extend the notion to the case of multiple classifiers, we can consider the output of such a randomized classifier to be a convex combination of the outputs of its endpoints Abnar et al. (2022). All the points on the convex hull are achievable. Therefore, the output of the randomized classifier is equivalent to the convex hull of our trained classifiers' performance.

In our experiments, we include the upper hull of the convex hull of the model performances, *i.e.*, the highest downstream performance for every given sample size. Such convex hull fit is proved to be robust to the density of the points in each figure Abnar et al. (2022).

A final remark is that our observations on different downstream datasets do not result in a one-model-fits-all trend. So we propose to fit a complicated curve whose function has form $f(x) = a_1 \ln x / x^{a_2} + a_3 \ (a_1, a_2, a_3 > 0)$ to the best performing models (*i.e.*, the convex hull fit as discussed above). The fitted parameters $a_1$, $a_2$ and $a_3$ in this function of each curve are given in Table 6.

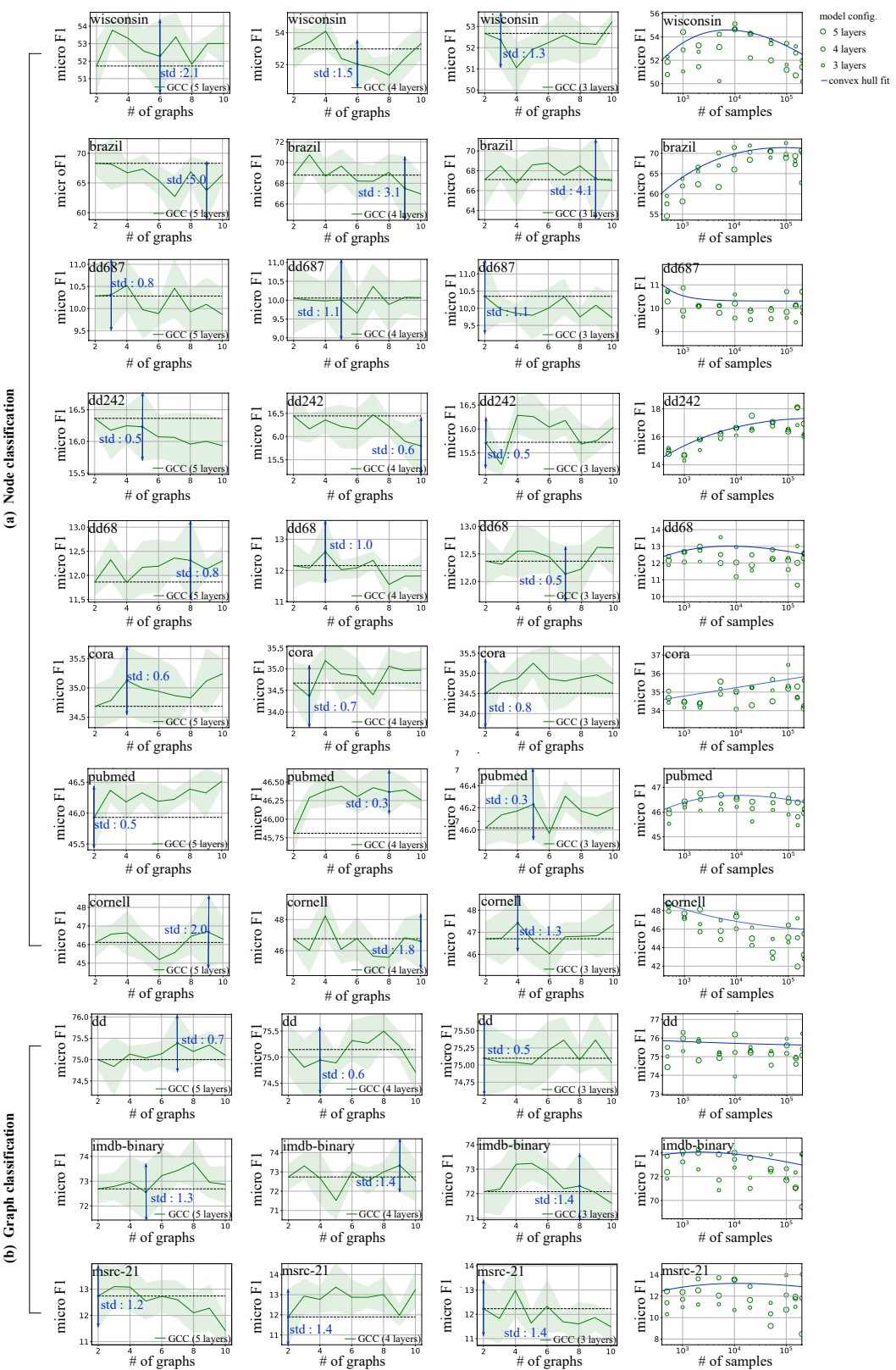

Figure 5: The additional observations of *curse of big data* phenomenon, performed on different GCC pre-training models.

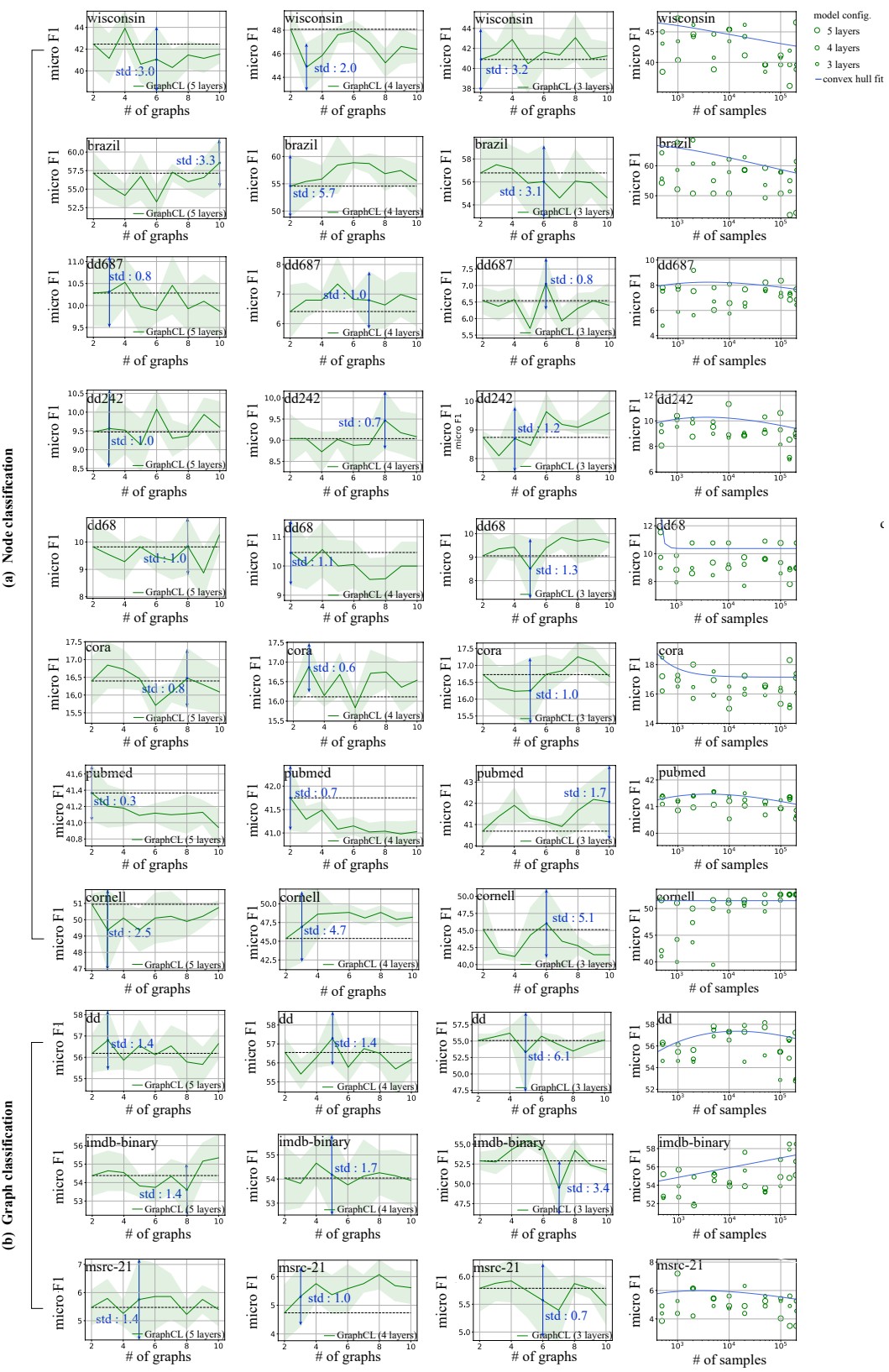

Figure 6: The additional observations of *curse of big data* phenomenon, performed on different GraphCL pre-training models.

# E EMPIRICAL STUDY OF GRAPH PROPERTIES

**Additional properties for part (b) in Figure 2.** In Figure 7, we plot the Pearson correlation between the graph properties of the graph used in pre-training (shown in the $y$-axis) and the performance of the pre-trained model using this graph on different unseen test datasets (shown in the $x$-axis). Note that the pre-training is performed on each of the input training graphs (in Table 3) via GCC. The results indicate that network entropy, density, average degree and degree variance exhibit a clear positive correlation with the performance, while the scale-free exponent presents an obviously negative relation with the performance. On the contrary, some other properties of graphs, including clique number, transitivity, degree assortativity and average clustering coefficient, do not seem to have connections with downstream performance, and also exhibit little or no correlation with the performance. Therefore, the favorable properties of network entropy, density, average degree, degree variance and the scale-free exponent of a real graph are able to characterize the contribution of a graph to pre-training.

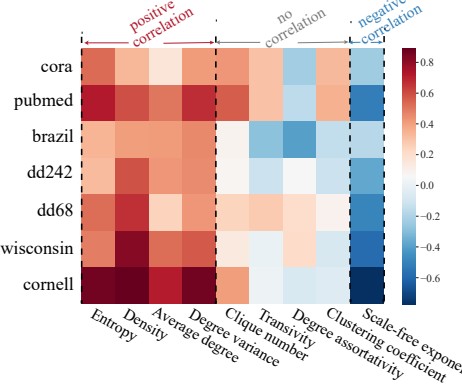

Figure 7: Pearson correlation between the structural features of the graph used in pre-training and the performance of the pre-trained model (using this graph) on different unseen test datasets.

**Detailed illustrations of Figure 3.** In Figure 3, the illustrative graphs are generated by the configuration model with 15-18 nodes. The shaded area groups the illustrative graphs whose network entropy and graph properties are similar. Each four points on the same horizontal coordinate represent four graph properties of an illustrating graph. Each curve is fitted by least squares and represents the relation between entropy and other graph properties.

**Additional real-world example for Figure 3.** In Figure 8, we provide a real-world example of how network entropy correlates with four typical structural properties (in red), as well as the performance of the pre-trained model on test graphs (in blue). Numerical experiments again support our explanation (or intuition) of their strong correlation.

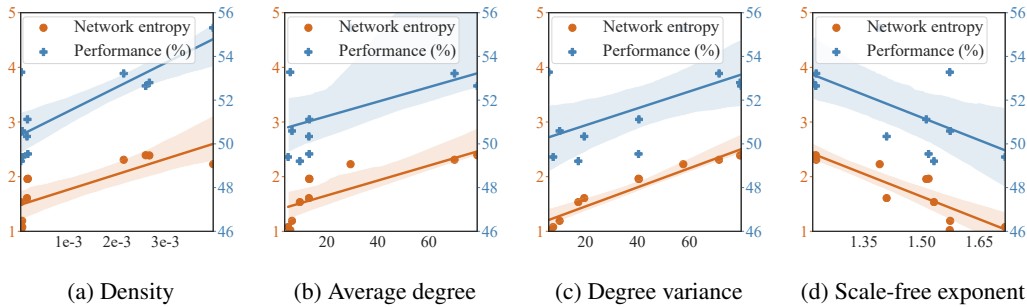

(a) Density      (b) Average degree      (c) Degree variance      (d) Scale-free exponent

Figure 8: The red plot shows the network entropy (left $y$-axis) versus typical structural properties in a graph (*i.e.*, density, average degree, degree variance, and the parameter $\alpha$ in a scale-free network), and the blue one shows the pre-training performances on wisconsin dataset (right $y$-axis) versus structural features.

# F    IMPLEMENTATION DETAILS

The number reported in all the experiments are the mean and standard deviation over 10 evaluation results of the downstream task with random training/testing splits. When conducting the downstream task, For each dataset, we consistently use 90% of the data as the training set, and 10% as the testing set. We conduct all experiments on a single machine of Linux system with an Intel Xeon Gold 5118 (128G memory) and a GeForce GTX Tesla P4 (8GB memory). Our codes are available at `https://github.com/anonymous-APT-ai/Anonymous-APT-code`.

**Implementations of our model.**    The regularization for weights of the model in Eq. (5) is applied to first 2 layers of GIN. The maximal period of training one graph $F$ is 6, the maximum iteration number $T$ is 100, and the predictive uncertainty thresholds $T_s$ and $T_g$ are set to be 3 and 2 respectively. The selected instances are sampled from 20,000 instances each epoch. Since the pre-training model is unable to provide precise predictive uncertainty in the initial training stage, the model is warmed up over the first 20 iterations. Since we adopt GCC as the backbone pre-training model, the other settings are the same as GCC.

Our model is implemented under the following software settings: Pytorch version 1.4.0+cu100, CUDA version 10.0, networkx version 2.3, DGL version 0.4.3post2, sklearn version 0.20.3, numpy version 1.19.4, Python version 3.7.1.

**Implementations of baselines.**    We compare against several graph representation learning methods. For implementation, we directly adopt their public source codes and most of their default hyperparameters. The key parameter settings and code links can be found in Table 7.

Table 7: The source code and major hyper-parameters used in the baselines.

| Method | Hyper-parameter | Code |
|---|---|---|
| DeepWalk | The dimension of output representations is 64, walk length = 10, number of walks = 80 | `https://github.com/shenweichen/GraphEmbedding` |
| struc2vec | The dimension of output representations is 32, walk length = 80, number of walks = 10, window size = 5 | `https://github.com/leoribeiro/struc2vec` |
| DGI | 512 hidden units per GNN layer, learning rate = 0.001 | `https://github.com/PetarV-/DGI` |
| GAE | 32 hidden units per GNN layer, learning rate = 0.01 | `https://github.com/zfjsail/gae-pytorch` |
| graph2vec | The dimension of output representations is 128 | `https://github.com/benedekrozemberczki/graph2vec` |
| InfoGraph | 32 hidden units per GNN layer, 5 layers | `https://github.com/fanyun-sun/InfoGraph` |
| DGCNN | 32 hidden units per GNN layer, learning rate = 0.001, batch size = 50 | `https://github.com/leftthomas/DGCNN` |
| GIN | 64 hidden units per GNN layer, 5 layers, learning rate = 0.01, sum pooling | `https://github.com/weihua916/powerful-gnns` |
| GraphCL | 300 hidden units per GNN layer, 5 layers, learning rate = 0.001 | `https://github.com/Shen-Lab/GraphCL` |
| GCC | 64 hidden units per GNN layer, 5 layers, learning rate = 0.005, number of samples per epoch = 20000 | `https://github.com/THUDM/GCC` |

## G    ADDITIONAL EXPERIMENTAL RESULTS

**Effects of hyper-parameter** $\{\lambda_j\}$**.**   The hyper-parameter $\{\lambda_j\}$ is the trade-off parameters between the knowledge learnt from new data and that from previous data in Eq. (5). We simply set $\lambda_1 = \lambda_2 = \cdots = \lambda_k$. We use the dataset dd242 as an example to find the suitable values of the hyper-parameter under the L2 and EWC regularization setting respectively, and present here for reference (see Figure 9). Clearly, a too small or too large $\lambda$ would deteriorate the performance. Thus, an appropriate value of $\lambda$ is preferred to ensure that the graph pre-training model can learn from new data as well as remember previous knowledge. We leave changing $\{\lambda_j\}$ as the future work.

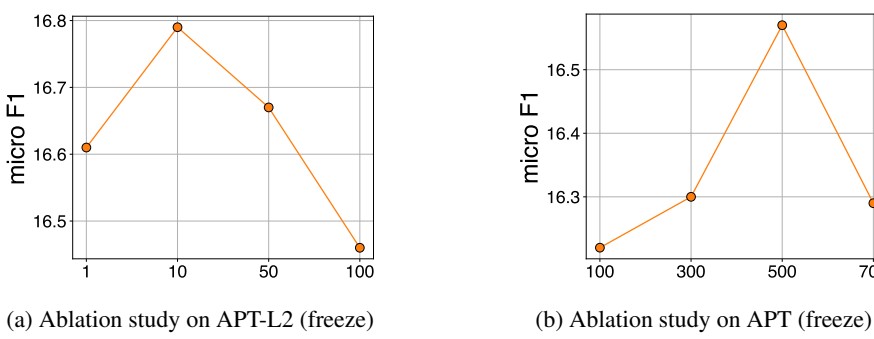

(a) Ablation study on APT-L2 (freeze)        (b) Ablation study on APT (freeze)

Figure 9: Performance of our model on dd242 w.r.t varying $\{\lambda_j\}$.

**Effects of hyper-parameter** $F, T_g, T_s$**.**        Our model training involves three hyper-parameter $F, T_g, T_s$, where $F$ controls the largest number of epochs training on each graph, $T_g$ is the predictive uncertainty threshold of moving to a new graph, $T_s$ is the predictive uncertainty threshold of choosing training samples. We use grid search to show $F \in \{4, 5, 6\}$'s, $T_g \in \{3, 3.5, 4\}$'s and $T_s \in \{1, 2, 3\}$'s role in the pre-training. $F$ remains at 5 while studying $T_g$ and $T_s$, $T_g$ remains at 3.5 while studying $F$ and $T_s$, and $T_s$ remains at 2 while studying $F$ and $T_g$. Figure 10 presents the effect of these parameters, We find that if the value of $F$ is set too small or that of $T_g$ is too large, the model cannot learn sufficient knowledge from each graph, leading to suboptimal results. Too large $F$ or small $T_g$ also lead to poor performance. This indicates that instead of training on a graph for a large period, it would be better to switch to training on various graphs in different domains to gain diverse and comprehensive knowledge. Regarding the hyper-parameter $T_s$, we observe that large $T_s$ would make the model having too few training samples to learn knowledge, and small $T_s$ could not select the most uncertain and representative samples, thus both cases achieve suboptimal performance.

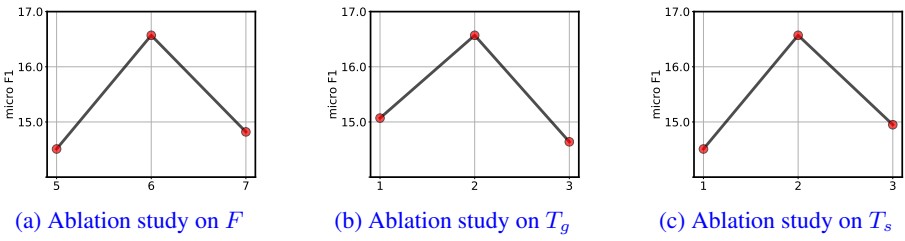

(a) Ablation study on $F$          (b) Ablation study on $T_g$          (c) Ablation study on $T_s$

Figure 10: Performance of our model on dd242 w.r.t varying $F, T_g, T_s$.

**The choice of** $\beta_t$**, its alternatives, and ablation study.**   At the beginning of the pre-training, the model is less accurate and needs more guidance from graph properties. We therefore set $\gamma_t$ as larger at the beginning and gradually decrease it. To simplify this process, we follow Cai et al. (2017) to use the exponential formula of $\beta_t = c_1 - c_2^t$ to set the expectation of $\gamma_t$ to be strictly decreasing (where $\gamma_t \sim \text{Beta}(1, \beta_t)$).

The parameters $c_1$ and $c_2$ in the exponential formula of $\beta_t = c_1 - c_2^t$ are suggested as 1.005 and 0.995 in Cai et al. (2017). We simply perform grid search on $c_1$ in $\{1.005, 3, 5\}$; see the effect of $c_1$ in the Figure 11.

Table 8: Micro F1 of APT-L2 (freeze) with the different decay functions in the node classification task.

| Method \ Dataset | brazil | dd242 | dd68 | dd687 | wisconsin | cornell | cora | pubmed |
|---|---|---|---|---|---|---|---|---|
| linear | **72.30(1.37)** | 16.28(0.57) | 12.44(0.72) | 10.29(0.87) | 54.20(1.50) | 47.66(1.53) | **35.74(0.52)** | 46.49(0.19) |
| step | 68.70(3.95) | 16.74(0.45) | **12.86(1.07)** | 10.09(0.76) | 52.55(2.39) | 48.08(1.28) | 35.50(0.46) | **46.58(0.21)** |
| exponential | 69.82(2.32) | **16.79(0.88)** | 12.68(0.81) | **10.34(1.12)** | **55.11(1.74)** | **48.76(2.20)** | 34.27(0.43) | 46.21(0.15) |

Table 9: Micro F1 of APT-L2 (freeze) with the different decay functions in the graph classification task.

| Method \ Dataset | imdb-binary | dd | msrc-21 |
|---|---|---|---|
| linear | **73.66(0.34)** | 75.47(0.26) | 13.01(0.78) |
| step | 72.99(0.40) | 75.41(0.41) | **14.13(0.56)** |
| exponential | 73.54(0.40) | **75.81(0.30)** | 13.16(0.77) |

We then illustrate that the choice of the decay function of $\beta_t$ is robust. Table 8 and Table 9 below show the effect of linear decay, step decay and exponential decay on $\beta_t$. (The function for linear decay and step decay are designed as $\beta_t = 2.001 + 0.004t$, $\beta_t = 2.005 + \text{floor}(t/20)$, respectively. The initial value $\beta_1$ is set the same as ours.) While there is no universally better decay function, the performance of our method is not significantly impacted by the choice of different decay functions, and our performance is better than the baselines in most cases regardless of the choices of specific decay functions.

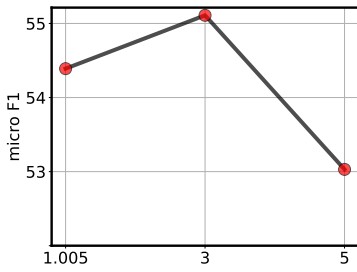
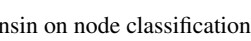
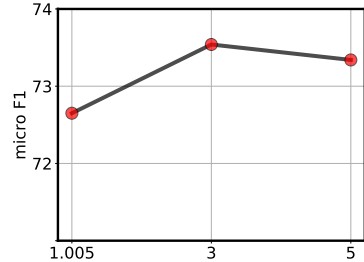

(a) Wisconsin on node classification      (b) Imdb-binary on graph classification

Figure 11: Performance of APT-L2 (freeze) w.r.t varying $c_1$.

**Impact of five graph properties combination.** As a further experimental analysis, we study the effect of the strategy of utilizing only one graph property in Table 10 and Table 11. We find that the five properties used in our model are all indispensable, and the most important one probably varies for different tasks and datasets. That's why we choose to combine all graph properties.

Moreover, these case studies may provide some clues of how to select pre-training graphs when some knowledge of the downstream tasks is known. For example, if the downstream dataset is extremely dense (like imdb-binary), the density property dominates among the selection criteria (such that the probability of encountering very dense out-of-distribution samples during testing can be reduced). If the entropy of downstream dataset is very high (like brazil), it is perhaps better to choose graphs with high entropy for pre-training. But still, when the downstream task is unknown, using the combination of five metrics often leads to the most satisfactory and robust results.

**The justification of input graphs' learning order.** Table 12 reveals the downstream performance can be affected by the learning order of input training graphs. With the guidance of graph selector, the pre-training model is encouraged to first learn the graphs and samples with higher predictive

uncertainty and graph properties. Such learning order accomplishes better downstream performance compared to the reverse or random one.

**The choice of the "difficult" data.** Among all the data, "difficult" samples contribute the most to the loss function, and thus they can yield gradients with large magnitude. Comparatively, training with easy samples may suffer from inefficiency and poor performance as these data points produce gradients with magnitudes close to zero Huang et al. (2016); Sohn (2016). In addition, learning from difficult samples has proven to be able to accelerate convergence and enhance the expressive power of the learnt representations Suh et al. (2019); Schroff et al. (2015). For our model, the importance of learning from difficult samples is also justified empirically, as shown in Table 13.

**Training time.** As empirically noted in Table 14, the total training time of APT-L2 and APT is 18321.39 seconds and 18592.01 seconds respectively (including the time consumed in graph selection and regularization term), while the competitive graph pre-training model GCC takes 40161.68 seconds for the same number of training epochs on the same datasets.

- The time spent on the inference on all graphs during graph selection (which is the main time spent for graph selection) only accounts for 3.95% and 3.87% of the total time under APT-L2 and APT respectively. Note that this step is executed only in a few epochs (around 6% in our current model) if and only if the condition in line 5 in Algorithm 1 is satisfied.

- The time cost of the L2 regularization term only accounts for 0.08% of the total time and the EWC regularization term only accounts for 0.45% of the total time, which is calculated by the runtime gap between the models with and without the regularization term. Note that the regularization term is imposed on the first two layers of the GNN encoder, which only accounts for 12.4% of the total number of parameters.

The efficiency of our model is due to a much smaller number of carefully selected training graphs and samples at each epoch. In addition, the number of parameters in our model is 190,544, which is the same order of magnitude as classical GNNs like GraphSAGE, GraphSAINT, *etc.* and is relatively small among models in open graph benchmark Hu et al. (2020b).

**Time comparison: pre-training vs. training from scratch.** Using a pre-trained model can significantly reduce the time required for training from scratch. The reason is that the weights of the pre-trained model have already been put close to appropriate and reasonable values; thus the model converges faster during fine-tuning on a test data. As shown in Figure 12, compared to regular GNN model (*e.g.* GIN), our model yields a speedup of 4.7× on average (which is measured by the ratio of the training time of GIN to the fine-tuning time of APT). Based on above analysis, we can draw a conclusion that pre-training is beneficial both in effectiveness and efficiency.

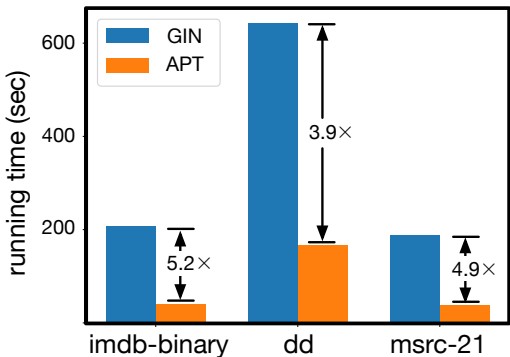

Figure 12: The running time of our model and the basic GNN model on graph classification task. Our model achieves a speedup of 4.7× on average compared with GIN.

Table 10: The effect of different graph properties on downstream performance (micro F1 is reported) under APT-L2 (fine-tune) in the node classification task. The last row is our strategy of combining all the graph properties, and each of the first five rows is the strategy of only utilizing one graph property.

| Method \ Dataset | brazil | dd242 | dd68 | dd687 | wisconsin | cornell | cora | pubmed |
|---|---|---|---|---|---|---|---|---|
| Entropy | **80.04(2.15)** | 25.79(0.94) | 16.31(0.81) | 11.08(0.82) | 67.01(2.00) | 52.80(2.46) | 45.41(0.85) | 50.85(0.19) |
| Density | 79.23(1.92) | **27.29(0.62)** | **19.89(0.95)** | 12.22(1.06) | 65.58(1.87) | 51.15(1.59) | 46.18(0.71) | 50.74(0.15) |
| Average degree | 79.22(1.65) | 24.99(0.67) | 16.56(1.01) | 11.67(1.18) | 67.02(1.86) | 51.43(4.16) | 46.38(0.48) | 50.99(0.31) |
| Degree variance | 78.44(2.24) | 24.94(0.61) | 16.62(1.04) | 11.51(1.17) | 65.65(1.28) | 50.45(2.14) | 45.76(0.65) | 50.70(0.21) |
| Scale-free exponent | 79.70(2.71) | 24.94(0.68) | 17.26(0.63) | 12.03(1.41) | 64.77(2.31) | 51.37(2.70) | 45.18(0.52) | 50.84(0.26) |
| Our combination | 78.75(1.63) | 24.62(0.90) | 17.83(1.35) | **12.26(0.78)** | **67.04(1.50)** | **52.94(1.95)** | **47.48(0.46)** | **51.25(0.21)** |

Table 11: The effect of different graph properties on downstream performance (micro F1 is reported) under APT-L2 (fine-tune) in the graph classification task.

| Method \ Dataset | imdb-binary | dd | msrc-21 |
|---|---|---|---|
| Entropy | 76.78(0.84) | 75.56(0.84) | 24.34(1.50) |
| Density | **77.20(0.66)** | 75.29(0.54) | 24.20(1.31) |
| Average degree | 76.87(0.93) | 75.46(0.53) | 25.14(1.54) |
| Degree variance | 76.39(1.04) | 75.47(0.67) | 25.22(1.51) |
| Scale-free exponent | 75.24(0.62) | 75.52(1.24) | 23.19(1.39) |
| Our combination | 75.93(0.84) | **75.58(1.06)** | **25.58(1.57)** |

Table 12: The effect of input graphs' learning order on downstream performance (micro F1 is reported) under freezing mode in the node classification task. The first row is the order learnt from APT-L2, and the second and third rows are the reverse and random order of the first row, respectively.

| Method \ Dataset | brazil | dd242 | dd68 | dd687 | wisconsin | cornell | cora | pubmed |
|---|---|---|---|---|---|---|---|---|
| Our order | **69.82(2.32)** | **16.79(0.88)** | **12.68(0.81)** | 10.34(1.12) | **55.11(1.74)** | **48.76(2.20)** | 34.27(0.43) | 46.21(0.15) |
| Reverse order | 69.60(2.71) | 16.00(0.47) | 11.41(0.91) | 10.65(0.65) | 51.46(1.64) | 44.36(1.38) | 35.66(0.62) | 45.92(0.14) |
| Random order | 67.25(2.40) | 16.11(0.79) | 12.57(1.17) | **11.06(0.75)** | 53.06(2.41) | 46.76(1.95) | **35.90(0.72)** | **46.36(0.20)** |

Table 13: The comparison of learning from easy samples and learning from difficult sample in our pipeline (APT-L2 (freeze)) on node classification. Micro F1 is reported in the table. (Under the setting of learning from easy samples, we replace $\phi_{\text{uncertain}}$ with $-\phi_{\text{uncertain}}$ in Eq.(4), and only sample instances with predictive uncertainty lower than $T_s$.)

| Method \ Dataset | brazil | dd242 | dd68 | dd687 | wisconsin | cornell | cora | pubmed |
|---|---|---|---|---|---|---|---|---|
| Learning from easy samples | 56.34(3.45) | 14.38(0.53) | 11.76(1.04) | 9.90(0.64) | 50.65(1.84) | 48.09(1.72) | **35.74(0.42)** | 46.03(0.17) |
| Learning from difficult samples (ours) | **69.82(2.32)** | **16.79(0.88)** | **12.68(0.81)** | **10.34(1.12)** | **55.11(1.74)** | **48.76(2.20)** | 34.27(0.43) | **46.21(0.15)** |

Table 14: Training time (sec) comparison between our model and GCC. All the models are trained under the same number of epochs, which is set as 100 in practice. (The difference in time cost of inference on all graphs is due to different runs.)

| | GCC | APT-L2 | APT |
|---|---|---|---|
| Time of the inference on all graphs | - | 723.92 | 719.64 |
| Time of the regularization term | - | 15.98 | 83.58 |
| **Total time** | 40161.68 | 18321.39 | 18592.01 |

## H   DISCUSSION: THE DESIGN OF PREDICTIVE UNCERTAINTY.

We here discuss two advantages of using the model loss (i.e., InfoNCE loss) as predictive uncertainty to select samples. First, InfoNCE loss is exactly the objective function of our model, so what we do is actually to select the samples with the greatest contributions to the objective function (i.e., select the samples with the greatest InfoNCE loss). Such strategy has been justified to accelerate convergence and enhance the discriminative power of the learned representations [1-4]. Second, as the loss function of our model, InfoNCE is already computed during the training, and thus no additional computation expense is needed in the data selection phase.

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
