# OpenReview forum: "Better with Less: Data-Active Pre-training of Graph Neural Networks"
_ICLR.cc/2023/Conference — Submitted to ICLR 2023_

### Official Review · Reviewer_pEvr · 2022-10-24

**Confidence:** 3
**Correctness:** 3
**Technical Novelty And Significance:** 2
**Empirical Novelty And Significance:** 2
**Recommendation:** 3

**Clarity, Quality, Novelty And Reproducibility:**

Clarity needs to be significantly improved;
Quality moderate;
Novelty limited;
Reproducibility good as code is provided.


**Strength And Weaknesses:**

Strength
1. The observation that more data does not necessarily leads to better accuracy is neat.
2. Using uncertainty and graph properties to select data makes sense.
3. The empirical results are strong.


Weakness
1. Discussions of related works are shallow, which make the contributions unclear. For instance, in Section 5, the Pre-training in CV and NLP part is nothing more than saying that CV and NLP have used pre-training. However, the readers are interested in what methods they use for pre-training, especially for selecting data, and what are the differences from this work. The same applies to the Graph pre-training part. My suggestion is that when you discuss some closely related works, you should make the readers understand how these works on the methodology level, instead of only describing their targets. To my knowledge, uncertainty is widely used for sample selection is many areas. What are the differences from this work?

2. Method design lacks explanation and does not make sense in some places. Eq. (4) calculates the mean of several graph property metrics (i.e., network entropy, density, degree). However, the appendix shows that these metrics differ in scale, which means that a metric may dominate the others when computing the mean. It also needs to be explained why we prefer larger values of these metrics (the network entropy part is clear). My guess that the intuition is that we prefer complex graphs that are likely to contain a diverse set of graph structures, and the proposed method tends to select graph with larger average degrees.

3. Experiment and analysis need to be significantly improved. (1) What is the time used for data selection? Will it be a concern? (2) In Tables 1-2, different variants of APR perform the best on different datasets, what are the reasons? Is it related to the properties of the test graph? (3) Why not compare with some well-known GNN baselines, such as GraphSAGE, which can be trained in an unsupervised manner? (4) How does different graph properties affect accuracy? How does the weight of uncertainty and graph properties affect performance?


**Summary Of The Paper:**

This paper proposes to select some graph data to improve the pre-training of GNNs. Existing works use all data to conduct pre-training, and it is observed that more data does not necessarily leads to better accuracy. The authors propose to use prediction uncertainty and graph properties to select data for training. Empirical results show that the proposed methods improve the accuracy of downstream tasks.

**Summary Of The Review:**

This paper proposes to select data to pre-train GNNs. The idea makes sense but there are significant problems in method design and experiment analysis. Novelty is also unclear from the presentation.

---

> ### Author Response · Authors · 2022-11-13
> **Response to Reviewer pEvr [5/5]**
>
> **Regarding the performance of different variants of APT.**
>
> First, on the node classification task (see Table 1), we analyze the performance of different variants from the following three points of view: (1) APT-L2 and APT, the variants with the proximity regularization w.r.t old knowledge, perform the best in most cases. This suggests that catastrophic forgetting of previous trained graphs could occur during pretraining, and it is necessary to add the proximity regularization to prevent this; (2) Besides, we observe that sometimes APT-R (the variant without proximity regularization) achieves the best performance on the test data dd687 and cornell. One potential reason is that the last data to be selected for pre-training is dblp, while the test data dd687 and cornell exhibit some properties similar to dblp. For example, the degree variance of dd687 and cornell is most similar to dblp in all test data, and the network entropy of dd687, cornell and dblp are all around 2. Therefore, even excluding the proximity regularization, the pre-training model can still remember knowledge from the last pre-trained data, and achieve good performance on the test data that is similar to the last pre-trained data; (3) One exception is that APT-G (the variant that removes the criteria of graph properties when selecting data) outperforms the other variants on dd242 under freezing mode, while the variants including graph properties and other components, APT, performs the best under fine-tuning. We find that dd242 is the test data with very small degree variance, average degree, network entropy, etc. among all test data, thus the variants selecting data with large value of these properties (variants exclude APT-G) would learn from the pre-training data that diverges the most from dd242, resulting in suboptimal performance. Nevertheless, these variants have rich knowledge, so they can adapt very well when further fine-tuned on test data and perform the best.
>
> For the graph classification task (see Table 2), we argue that different variants of APT perform similarly, and it is difficult to tell which one is significantly better than the others. (But APT outperforms other baselines in a noticeable manner.) We suspect that this phenomenon is more related to the task rather than the test graph. The underlying reason might be that the backbone pre-training model GCC is a kind of node-level contrastive learning, so the components we design in the paper may not necessarily show the distinction on graph-level tasks.
>
> **Regarding the comparison with well-known GNN baselines.**
>
> Thanks for the suggestion. We have added the unsupervised GraphSAGE as our baselines under node classification in Table 1. (The graph classification cannot be adapted and implemented in unsupervised GraphSAGE.) The results show that our model consistently outperforms GraphSAGE, which indicates our generalization ability.
>
> | Dataset|  brazil | dd242| dd68|dd687|wisconsin|cornell|cora | pubmed   | ogbarxiv|ogbproteins|
> |---------|:----:|:-----:|:-----:|:------:|:-----:|:------:|:-----:|:------:|:------:|:-----:|
> | GraphSAGE   |         67.93(9.28)        |         14.33(0.37)        |         13.55(0.70)        |         10.39(0.78)        |         47.03(1.98)        |        49.20(1.46)       |       35.93(1.76)      |       39.94(0.02)      |      /          |           /          |
> | APT (freeze)    |   73.39(2.55)   |        16.57(0.94)       |        12.08(0.89)       |   10.35(1.24)       |        53.38(1.19)       |       47.37(1.29)      | 36.69(0.49) | 46.88(0.21) | 22.04(0.29) | 62.29(0.55) |
> | APT (fine-tune)      | 79.67(2.30) | 28.62(0.55) | 20.30(1.13) | 12.80(1.54) | 67.08(1.75) |       52.15(2.25)      | 47.51(0.62) | 51.30(0.16) | 27.40(4.87) | 61.64(0.35) |
>
> Table:  Micro F1 scores of GraphSAGE and our model in the node classification task.
>
> **Regarding the effect of different graph properties and the combination weight.**
>
> The effect of different graph properties is reported in Table 10 and Table 11 in Appendix G. Ablation study finds that the five properties used in our model are all indispensable, and the most important one varies for different tasks and datasets. That’s why we choose to combine all graph properties.
>
> The weight of uncertainty and graph properties is also analyzed in Appendix G, under the title "The choice of β_t , its alternatives, and ablation study". We show that the choice of the weight's decay function is robust in our model, and study parameter sensitivity in the combination weight.

---

> ### Author Response · Authors · 2022-11-13
> **Response to Reviewer pEvr [4/5]**
>
> > Q2: Method design lacks explanation and does not make sense in some places. Eq. (4) calculates the mean of several graph property metrics (i.e., network entropy, density, degree). However, the appendix shows that these metrics differ in scale, which means that a metric may dominate the others when computing the mean.  It also needs to be explained why we prefer larger values of these metrics (the network entropy part is clear). My guess that the intuition is that we prefer complex graphs that are likely to contain a diverse set of graph structures, and the proposed method tends to select graph with larger average degrees.
>
> First, we would like to clarify that each loss term is the z-score normalized value, which is insensitive to the scale of graph properties and graph-level predictive uncertainty.
>
> Second, the motivations for c density, average degree, degree variance and scale-free exponent are attached in the appendix (due to space limit), which exactly coincide with your guess of the reason of diversity.  As explained in the last paragraph Appendix A: "graphs with larger average degree and higher density have more interactions among the nodes, thus providing more topological information to graph pre-training. Also, the larger the diversity of node degrees, the more diverse the subgraph samples. The diversity of node degrees can be measured by degree variance and scale-free exponent. (A smaller scale-free exponent indicates the length of the tail of degree distribution is relatively longer, i.e., the degree distribution spreads out wider. )". Besides, in Figure 7 and Figure 8 in Appendix E, we also provide the empirical justification of the strong correlation between these properties of the graph used in pre-training and the performance of the pre-trained model (using this graph) on different unseen test datasets. We have mentioned these analyses in the last paragraph of "Graph properties" in Section 3.1.
>
> > Q3: Experiment and analysis need to be significantly improved. (1) What is the time used for data selection? Will it be a concern? (2) In Tables 1-2, different variants of APR perform the best on different datasets, what are the reasons? Is it related to the properties of the test graph? (3) Why not compare with some well-known GNN baselines, such as GraphSAGE, which can be trained in an unsupervised manner? (4) How does different graph properties affect accuracy? How does the weight of uncertainty and graph properties affect performance?
>
> **Regarding the time used for data selection.**
>
> Thank you for pointing out the time concern. The computation time is reported in Appendix G (due to page limit). The time used for data selection mainly comes from the computation of the predictive uncertainty, which is exactly the time spent on the inference on all graphs during data selection. The time regarding graph properties is negligible, because they are calculated before pre-training. In our experiment, data selection takes 719.64 seconds and only accounts for 3.87% of the total time of APT.

---

> ### Author Response · Authors · 2022-11-13
> **Response to Reviewer pEvr [3/5]**
>
> **Regarding the related works of uncertainty-based sample selection.**
>
> Thanks for pointing out this. We discussed the related works of uncertainty-based sample selection as follows.
>
> Uncertainty is a widely-used terminology in machine learning, and does not have a clear/universally-accepted definition. In general, this word is used to refer to lack of confidence for a parameter/measure/quantity in a machine learning model/algorithm. One line of works in active learning introduce the measure of uncertainty by querying the labels of samples that current model is least certain w.r.t classification prediction, and choosing the samples with large uncertainty to label (Cai et al. (2017); Zhang et al. (2021a); Yang et al. (2015); Zhu et al. (2008)). However, such uncertainty measure relies on the labels, which cannot be adapted in pre-training with unlabeled data. Another line of works in hard example mining introduce some similar strategies of selecting samples with greatest loss, which can also be regarded as a kind of uncertainty (Simo-Serra et al. (2014); Loshchilov & Hutter (2015)). However, these works only select i.i.d. samples with the greatest loss, while the predictive uncertainty in our manuscript is used to select not only samples but also pre-training graphs. Moreover, hard example mining fails to select samples and graphs with the following two requirements. (1) The chosen samples should follow a joint distribution that reflects the topological structures of real-world graphs. (2) The chosen set of graphs should include informative and sufficiently diverse samples. These two requirements, unachievable by hard example mining, are critical in our scenario. The first requirement is met by the use of graph-level predictive uncertainty and graph properties. (Note that the graph-level predictive uncertainty is different from the predictive uncertainty of a sample.) Then the second goal is achieved by the data-active graph pre-training framework, which provides the interaction between the graph selector and pre-training model.
>
> The connection and difference of our strategy with hard example mining have been discussed in the last paragraph of Section 3.1. So we additionally include the discussion of the uncertainty introduced in active learning in the last paragraph of Section 3.1 in the revised manuscript.

---

> ### Author Response · Authors · 2022-11-13
> **Response to Reviewer pEvr [2/5]**
>
> **Regarding the related works in graph pre-training.**
>
> Thanks for pointing out this. We have detailedly discussed the works in graph pre-training from methodology level as follows, and included in the revised version.
>
> Taking inspiration from the pre-training in CV and NLP, recent efforts have shed the light on pre-training GNNs. Initially, some classical unsupervised graph representation learning can be used for graph pre-training (Tang et al. (2015); He et al. (2016); Grover & Leskovec (2016); Narayanan et al. (2017); Ribeiro et al. (2017); Donnat et al. (2018); Zhang et al. (2019); Hamilton et al. (2017)). These models are designed based on neighborhood similarity assumption, which makes these models cannot generalize to unseen nodes and graphs. Later, a line of graph self-supervised learning can be also treated as graph pre-training, which can be categorized into two folds: graph generative models and contrastive models. Graph generative models aim to capture the universal graph patterns by recovering certain parts of the input graph (such as masked structure or attributes) (Kipf et al. (2016); Wang et al. (2017); Hu et al. (2020c); Cui et al. (2020); Hou et al. (2022)). Representative works include autoregressive models like GPT-GNN  (Hu et al. (2020c)) and autoencoders such as GAE  (Kipf et al. (2016)), MGAE  (Wang et al. (2017)), AGE  (Cui et al. (2020)), GraphMAE  (Hou et al. (2022)). These works rely on specific domain knowledge, for example the node/edge/attribute type should be the same, which makes generative models difficult to transfer across different types of graphs. On the other hand, graph contrastive models learn discriminative representations by maximizing the agreement between positive pairs and minimizing that between negative pairs  (Velickovic et al. (2019); Hu et al. (2020b); You et al. (2020a); Zhu et al. (2020); Hassani & Khasahmadi (2020); Sun et al. (2020); Li et al. (2021); Lu et al. (2021); Sun et al. (2021); Zhu et al. (2021b); Xu et al. (2021); Zhu et al. (2021a); Lee et al. (2022); Zeng & Xie (2021); Zhang et al. (2021b); Han et al. (2022)). One of the technical cores is to design appropriate graph data augmentation strategies, including attribute masking, edge perturbation, node dropping, diffusion, etc. These augmentations are either performed on node attributes or the whole graph structure, so they only achieve transferability in graphs from similar (or the same) domains, or the downstream task is restricted to graph classification. With the purpose of learning transferable graph patterns across different domains, some researchers take subgraph sampling as an augmentation strategy, such that the transferable (sampled) subgraph patterns can be captured during pre-training (Qiu et al. (2020); You et al. (2020a; 2021)). However, the existing works on graph pre-training only focus on how to design the pre-training model, rather than how to select data for pre-training. Our paper first points out the necessity of selecting data, and fills the gap of the data selection strategy in graph pre-training.

---

> ### Author Response · Authors · 2022-11-13
> **Response to Reviewer pEvr [1/5]**
>
> Dear reviewer pEvr,
>
> We really appreciate your valuable comments. We have followed closely the suggestions, and made clarifications and revisions accordingly. We hope the newly provided content could help to further strengthen our work.
>
> > Q1: Discussions of related works are shallow, which make the contributions unclear. For instance, in Section 5, the Pre-training in CV and NLP part is nothing more than saying that CV and NLP have used pre-training. However, the readers are interested in what methods they use for pre-training, especially for selecting data, and what are the differences from this work.  The same applies to in the Graph pre-training part. My suggestion is that when you discuss some closely related works, you should make the readers understand how these works on the methodology level, instead of only describing their targets.  To my knowledge, uncertainty is widely used for sample selection is many areas. What are the differences from this work?
>
> Thanks for pointing it out. We first summarize our contributions, and then provide a more comprehensive discussion on related works. The discussion has also been added to Section 5 and the last paragraph in Section 3.1 in the revised manuscript. (The references mentioned in this response are numerous and we omit them here. You can find them in the revised manuscript.)
>
> Our work makes the following contributions:
> 1. To the best of our knowledge, our paper is the first one to reveal and study the curse of big data phenomenon in graph pre-training. This surprising phenomenon is only observed in the graph pre-training field; for pre-training in CV and NLP, more training data does improve the downstream performance (until some kind of saturation occurs).
> 2. In view of this phenomenon, we propose a novel graph pre-training framework to achieve better downstream performance with carefully chosen data. The problem of how to wisely choose suitable graphs and samples for graph pre-training has never been addressed in the existing literature. To fill this gap, we introduce a novel graph selector based on two criteria: predictive uncertainty and graph properties, and integrate the graph selector and the pre-training model into a unified framework such that they can actively cooperate with each other.
>
> **Regarding the related works of pre-training in CV and NLP.**
>
> Thanks for reminding us of these related works. We have detailedly discussed them as follows, and included in the revised version.
>
> Pre-training has received significant success in the fields of computer vision (CV) and natural language processing (NLP). Initially, the CV community benefits from the models like Vision Transformers (Liu et al. (2021)),  MLP-mixers   (Tolstikhin et al. (2021)) and ResNets  (He et al. (2016)), which are pre-trained on a large amount of image data in a supervised fashion. To take full advantage of large-scale unlabeled data, NLP community adapts self-supervised learning to leverage intrinsic information in text data. For example, GPT (Radford & Narasimhan (2018))  and BERT (Devlin et al. (2019)) are proposed for NLP tasks to learn a Transformer encoder (Vaswani et al. (2017)) from a large corpus. The success of these pre-training methods is attributed to the ubiquitous, transferable abstraction across the training and testing data in CV/NLP. However, such a property no longer exists in graph data, where graphs from different domains have numerous, diverse structures.
>
> Typically, for pre-training in CV and NLP, the intuition is that the more the pre-training data, the better the performance of the pre-trained model (Kaplan et al. (2020), Bello et al. (2021),Tan & Le (2019)). Later, a contrary viewpoint argues that in some cases, scaling up the pre-training data size results in a saturating performance or limited improvement in downstream. This phenomenon has been verified both in CV tasks (like image recognition) (Abnar et al. (2022); El-Nouby et al. (2021)) and NLP tasks (like text classification) (Raffel et al. (2020)). However, this is not true in graph pre-training. In this paper we argue that adding input graphs or pre-training samples does not necessarily improve, but sometimes even deteriorates the downstream performance.
>
> In view of the above phenomenon in CV and NLP pre-training, data selection is not an active research direction. The only related research we notice focuses on domain-specific pre-training models (Cui et al. (2018); Beltagy et al. (2019); Dai et al. (2019; 2020); Yan et al. (2020); Lee et al. (2020); Chakraborty et al. (2022)), which select pre-training data in the same domain as the downstream. The assumption on the knowledge of the downstream domain is different from the across-domain graph pre-training in our paper, and thus data selection in CV/NLP pre-training is not that relevant to the current work.

---

> ### Author Response · Authors · 2022-11-24
> **Any unanswered question yet?**
>
> Dear Reviewer pEvr,
>
> Thanks again for your detailed review and questions. We have provided detailed answers to them in the rebuttal and would like to kindly ask if you still have any unanswered questions about our paper?
>
> Best,
> Authors

---

> ### Comment · Area_Chair_raJQ · 2022-12-13
> **Notice**
>
> Dear Reviewer,
>
> If you do not want to update your score, please at least acknowledge that you have already read the rebuttal.

---

### Official Review · Reviewer_z7Ev · 2022-10-28

**Confidence:** 3
**Clarity, Quality, Novelty And Reproducibility:** See above comments.
**Correctness:** 3
**Technical Novelty And Significance:** 2
**Empirical Novelty And Significance:** 3
**Recommendation:** 6

**Strength And Weaknesses:**

Strengths:
1. The problem of GNN pre-training is drawing growing attention in the graph machine learning field. Most works for GNN pre-training aim to collect more data or design better pre-training objectives to boost the performance. By contrast, this paper proposes a new idea, i.e., using data in a more efficient way by picking up those informative examples for training. Overall, I think the idea is interesting and novel.
2. The proposed approach is quite intuitive.
3. The experiment is extensive, where 13 graphs are used for pre-training. The results show that the proposed approach APT outperforms many existing methods.

Weaknesses:
1. In Sec. 3.1, this paper defines the predictive uncertainty by using the InfoNCE loss. Although this definition is intuitive, I feel like the design is heuristic and is lack of theoretical guarantee. It would be helpful to further discuss the advantage of the InfoNCE loss for measuring the sample importance.
2. In terms of graph properties, this paper considers a few features of graphs, including network entropy, density, average degree, and, degree variance. Although they are very important to describe a graph, I feel like these properties are insufficient to yield a comprehensive characterization of the graph. I wonder whether it is possible to automatically learn some features as the structural properties of graphs.

**Summary Of The Paper:**

This paper focuses on pre-training graph neural networks (GNNs), aiming to identify a few most important examples for pre-training. For this purpose, a graph selector is designed, which uses the predictive uncertainty from a pre-trained GNN model as features for data selection. Extensive experiments on a few datasets prove the effectiveness of the proposed framework.

**Summary Of The Review:**

See above comments.

---

> ### Author Response · Authors · 2022-11-13
> **Response to Reviewer z7Ev**
>
> Dear reviewer z7Ev,
>
> We really appreciate your insightful comments. We have followed closely the suggestions, and made clarifications and revisions accordingly. Below is a point-by-point response.
>
> > Q1: In Sec. 3.1, this paper defines the predictive uncertainty by using the InfoNCE loss. Although this definition is intuitive, I feel like the design is heuristic and is lack of theoretical guarantee. It would be helpful to further discuss the advantage of the InfoNCE loss for measuring the sample importance.
>
> Thanks for the valuable comment. We here discuss two advantages of using the model loss (i.e., InfoNCE loss) to select samples. First, InfoNCE loss is exactly the objective function of our model, so what we do is actually to select the samples with the greatest contributions to the objective function (i.e., select the samples with the greatest InfoNCE loss). Such strategy has been justified to accelerate convergence and enhance the discriminative power of the learned representations [1-4]. Second, as the loss function of our model, InfoNCE is already computed during the training, and thus no additional computation expense is needed in the data selection phase. We have included the discussion in Appendix H in the revised version.
>
> > Q2: In terms of graph properties, this paper considers a few features of graphs, including network entropy, density, average degree, and, degree variance. Although they are very important to describe a graph, I feel like these properties are insufficient to yield a comprehensive characterization of the graph. I wonder whether it is possible to automatically learn some features as the structural properties of graph.
>
> We thank the reviewer for this meaningful and insightful comment. The graph properties we have explored in our work were originated from different research areas and proposed for various applications. It is difficult for the graph theory community to develop a systematic approach to study all these properties. With the advances of deep learning, it may be possible to explore and design more complicated and useful structural properties. The current work might benefit from advances in this direction. We sincerely believe that the reviewer's idea would be a promising yet largely unexplored research topic for both graph representation learning and machine learning explainability.
>
> **Reference:**
>
> [1] Suh, Yumin, Bohyung Han, Wonsik Kim and Kyoung Mu Lee. 2019. Stochastic Class-Based Hard Example Mining for Deep Metric Learning. In CVPR, pages 7244-7252.
>
> [2] Florian Schroff, Dmitry Kalenichenko, and James Philbin. 2015. FaceNet: A unified embedding for face recognition and clustering. In CVPR, pages 815-823.
>
> [3] Ilya Loshchilov and Frank Hutter. Online batch selection for faster training of neural networks. 2015. arXiv preprint arXiv:1511.06343, 2015.
>
> [4] Edgar Simo-Serra, Eduard Trulls, Luis Ferraz, Iasonas Kokkinos, and Francesc Moreno-Noguer. Fracking deep convolutional image descriptors. 2014. arXiv preprint arXiv:1412.6537, 2014.

---

> > ### Comment · Reviewer_z7Ev · 2022-11-24
> > **Response**
> >
> > Thanks the authors for the helpful feedback, which addressed most of my concerns, and I will keep the score of a weak accept.

---

### Official Review · Reviewer_xmXz · 2022-10-31

**Confidence:** 4
**Correctness:** 3
**Technical Novelty And Significance:** 3
**Empirical Novelty And Significance:** 3
**Recommendation:** 8

**Clarity, Quality, Novelty And Reproducibility:**

Clarity: The motivation, explanation, and results are clear. The clarity of the experiment section could be improved with reduced reliance on the Appendix.

Quality: The paper is well-written, and the results support the claims.

Novelty: Some of the ideas are present in the existing literature, but their implementation in the graph pretraining setting is new. The writing does a good job of placing the work appropriately with respect to existing literature.

Reproducibility: An anonymous code repository has been provided.


**Strength And Weaknesses:**

Strengths:

- The paper is overall well presented, and the methods are well described
- It aims to tackle an important problem in the field and proposes an interesting strategy to do so
- The experiments and results are thorough and demonstrate better downstream performance on a variety of tasks with fewer training samples
- The presentation of the phenomenon of large training samples not adding to the performance is useful

Weakness:

It would  be helpful if the following points regarding experimental settings, results, and implementation could be clarified:

- Why were different models used for node (logistic regression) versus graph classification (SVMs)?
- The ProNE model seems to outperform the APT model significantly for certain datasets. It would be helpful to discuss this result.
- How is F (the maximal period of training one graph) different from the training epochs, or is F measured in seconds?
- How sensitive is the model to the predictive uncertainty thresholds Tg and Ts (set to 3 and 2, respectively)? The decision choices for F, Tg, and Ts could be better described.
- Are the baseline methods being re-trained on new datasets? If so, does it make sense to use the default hyperparameters for the same?

Minor Comments:
- I may have missed this, but what was the selection strategy for pre-training versus test data?
- In appendix C, Table 3, the description of the ‘msu’ datasets implies it belongs to the protein category instead of the social category. The authors should make corrections where necessary.
- In appendix C, specifically in Tables 3,4, and 5, it would be helpful to define the abbreviations |V| , |E|, and |G| in the captions.


**Summary Of The Paper:**

This paper introduces a novel framework for cross-domain graph pre-training using fewer training samples. The paper first demonstrates the phenomenon of the “curse of big data” - more training and graph datasets do not always lead to better performance in downstream node and graph classification tasks. Next, it presents a data-active graph pre-training (APT) framework that consists of two parts.  (1) The graph selector uses a loss that combines the predictive uncertainty of a data sample and its graph properties (entropy, density, etc.) to make an informed decision on the most instructive data samples for the model. (2) The pre-training model that learns from the graph selector and provides it guidance for better selection.  The paper implements the framework for different datasets and tasks to show that the proposed method improves performance for downstream tasks in different domains. The proposed method is compared with existing graph-based pre-trained models. The paper also includes results on the ablation of the different components of the graph selector, which is an important analysis for such a framework.

**Summary Of The Review:**

The studied problem in this paper is important, the idea is new, and the experimental results are comprehensive. Specifically, it is demonstrated that the APT framework can achieve SOTA performance on downstream tasks by using less pre-training cross-domain data. The paper also shows this framework reduces the time complexity for pre-training. It would be great if some of the experimental choices and results could be described more clearly.

---

> ### Author Response · Authors · 2022-11-13
> **Response to Reviewer xmXz [2/2]**
>
> > Q5: Are the baseline methods being re-trained on new datasets? If so, does it make sense to use the default hyperparameters for the same?
>
> Sorry for the unclear description. The baseline methods are re-trained on new datasets. When implementing the baselines, we follow the default settings of most hyper-parameters. This is because we found these hyper-parameters in their original paper to be insensitive to different datasets. Besides, such strategy is usually taken as a common practice, which is also adopted by a set of graph pre-training works [1-3]. Nonetheless, we do think it is more appropriate to find the optimal hyper-parameters for the baselines. Now we try to perform grid search on the important hyperparameter hidden size on several baselines. The table below shows that the baselines with the default hidden size used in our paper either achieves the best performance or are close to the best performance in most cases. For the other baselines, we are working on them and wish to report the results in near future.
>
> | Method | hidden size | DD68        |
> |--------|-------------|-------------|
> | DGI    | 128         | 8.91(3.25)  |
> |        | 256         | 11.48(2.31) |
> |        | 512         | 13.57(0.44) |
> | GAE    | 16          | 12.85(0.80) |
> |        | 32          | 13.43(0.96) |
> |        | 64          | 13.29(0.90) |
> | ProNE      | 64          | 6.83(3.34)  |
> |            | 128         | 7.73(3.11)  |
> |            | 256         | 7.56(2.87)  |
> | Deepwalk   | 32          | 6.94(3.22)  |
> |            | 64          | 6.72(3.04)  |
> |            | 128         | 6.61(2.98)  |
> | struc2vec | 16          | 9.97(2.56)  |
> |            | 32          | 10.30(3.23) |
> |            | 64          | 10.48(2.31) |
>
> > Q6: I may have missed this, but what was the selection strategy for pre-training versus test data?
>
> Thanks for pointing it out. We here illustrate the criterion of selecting the pre-training data and test data. Overall, the selected pre-training data and test data should cover a wide spectrum of domains. The consideration of the graphs for pre-training and test is as follows. When selecting pre-training data, we hope that the graph size is at least hundreds of thousands to contain enough information for pre-training. When selecting test data, we hope that: (1) some test data is in the same domain as the pre-training data, and some is cross-domain, so as to comprehensively evaluate our model’s in and across-domain transferability. Accordingly, the in-domain test data is selected from the type of movie and citations, and the other test data are across-domain; (2) the size of test graphs can scale from hundreds to millions. The selection criterion has been included in Appendix C in the revised version.
>
> > Q7: In appendix C, Table 3, the description of the ‘msu’ datasets implies it belongs to the protein category instead of the social category. The authors should make corrections where necessary.
>
> Thanks for pointing out the mistake. We have corrected the description of msu dataset as the friendships between Facebook users in Michigan State University in the revised version, which belongs to the social category.
>
> > Q8: In appendix C, specifically in Tables 3, 4, and 5, it would be helpful to define the abbreviations |V| , |E|, and |G| in the captions.
>
> Thanks for pointing it out. We have added the definitions of abbreviations |V| , |E|, and |G| (i.e., the number of nodes in a graph, the number of edges in a graph and the number of graphs in graph classification datasets, respectively) in the caption of Table 3, 4 and 5 in our revised version.
>
> **Reference**
>
> [1] Jiezhong Qiu, Qibin Chen, Yuxiao Dong, Jing Zhang, Hongxia Yang, Ming Ding, Kuansan Wang, and Jie Tang. 2020.
> Gcc: Graph contrastive coding for graph neural network pre-training. In SIGKDD, pages 1150–1160.
>
> [2] Yuning You, Tianlong Chen, Yongduo Sui, Ting Chen, Zhangyang Wang, and Yang Shen. 2020a. Graph contrastive learning with augmentations. In NeurIPS, pages 5812–5823.
>
> [3] Zhenyu Hou, Xiao Liu, Yukuo Cen, Yuxiao Dong, Hongxia Yang, Chunjie Wang, and Jie Tang. 2022. Graphmae: Self-supervised masked graph autoencoders. In SIGKDD, pages 594–604.

---

> ### Author Response · Authors · 2022-11-13
> **Response to Reviewer xmXz [1/2]**
>
> Dear reviewer xmXz,
>
> We really appreciate your comments and your support for our work. We have followed closely the suggestions, and made clarifications and revisions accordingly. Below is a point-by-point response.
>
> > Q1: Why were different models used for node (logistic regression) versus graph classification (SVMs)?
>
> Sorry for the unclear description. For different classification tasks, we follow the settings of most baselines used in each task. For example, baselines including GCC, struc2vec, DeepWalk, ProNE, DGI and GraphSAGE perform node classification with logistic regression and baselines including GCC, graph2vec, InfoGraph, GIN, DGCNN, GraphCL, JOAO and GraphMAE perform graph classification with SVM. We have clarified this in Section 4 in the revised version.
>
> > Q2: The ProNE model seems to outperform the APT model significantly for certain datasets. It would be helpful to discuss this result.
>
> Thanks for mentioning this point. Actually, ProNE is a proximity-based model, which enforces neighboring nodes share similar representations. So ProNE would perform well on graphs with strong homophily (i.e., most connections happen among nodes in the same class) like cora and pubmed, while it is not suitable for the graphs with weak homophily like dd242 and cornell. What's more, it should be noted that the proximity information is not transferable: it is only useful when learning from one graph, but fails in the pre-training setting across different graphs. We have illustrated the discussion in Section 4.2 in the revised manuscript.
>
> > Q3: How is F (the maximal period of training one graph) different from the training epochs, or is F measured in seconds?
>
> Sorry for the misunderstanding. Our model trains several carefully selected graphs one-by-one in a sequential manner. For each selected graph, our model trains on it for at most F training iterations, while the training epochs T indicates the total number of iterations required to complete the training on all selected graphs. We have illustrated this in Appendix B in the revised manuscript.
>
> > Q4: How sensitive is the model to the predictive uncertainty thresholds Tg and Ts (set to 3 and 2, respectively)? The decision choices for F, Tg, and Ts could be better described.
>
> Thanks for reminding us of this analysis. We have provided detailed analysis of the choices of F, Tg, and Ts in Appendix G, under the title "Effects of hyper-parameter F, Tg , Ts", in our revised manuscript. The following table presents the effect of these parameters (or refer to Figure 10 in the Appendix for a clearer illustration).
>
> More specifically, among these three hyper-parameters, F controls the largest number of epochs training on each graph, Tg is the predictive uncertainty threshold of moving to a new graph, Ts is the predictive uncertainty threshold of choosing training samples. We use grid search to show F ∈ {4, 5, 6}’s, Tg ∈ {3, 3.5, 4}’s and Ts ∈ {1, 2, 3}’s role in the pre-training. F remains at 5 while studying Tg and Ts, Tg remains at 3.5 while studying F and Ts, and Ts remains at 2 while studying F and Tg. We find that if the value of F is set too small or that of Tg is too large, the model cannot learn sufficient knowledge from each graph, leading to suboptimal results. Too large F or small Tg also leads to poor performance. This indicates that instead of training on a graph for a large period, it would be better to switch to training on various graphs in different domains to gain diverse and comprehensive knowledge. Regarding the hyper-parameter Ts, we observe that large Ts would make the model have too few training samples to learn knowledge, and small Ts could not select the most uncertain and representative samples, thus both cases achieve suboptimal performance.
> | F        | 5           | 6           | 7           |
> |----------|-------------|-------------|-------------|
> | mirco-F1 | 14.51(0.07) | 16.57(0.94) | 14.82(0.19) |
>
> Table: Performance of APT (freeze) on dd242 w.r.t varying F.
>
> |  T_s     | 1           | 2           | 3         |
> |----------|-------------|-------------|-------------|
> | mirco-F1 | 14.51(0.58) | 16.57(0.94) | 14.95(0.64) |
>
> Table: Performance of APT (freeze) on dd242 w.r.t varying Ts.
>
> |  T_g     | 3           | 3.5         | 4          |
> |----------|-------------|-------------|-------------|
> | mirco-F1 |15.07(0.19) | 16.57(0.94) | 14.64(0.31) |
>
> Table: Performance of APT (freeze) on dd242 w.r.t varying Tg.

---

> > ### Comment · Reviewer_xmXz · 2022-11-29
> > **Response to authors**
> >
> > Thank you to the authors for addressing all my points. I will retain my original high score.

---

### Official Review · Reviewer_ajow · 2022-10-31

**Confidence:** 4
**Clarity, Quality, Novelty And Reproducibility:** Please refer to Strength And Weaknesses.
**Correctness:** 3
**Technical Novelty And Significance:** 3
**Empirical Novelty And Significance:** 2
**Recommendation:** 3

**Strength And Weaknesses:**

**Strength**

- Overall, the problem studied seems to be a novel angle in the context of GNN pre-training. Efficient pre-training is currently a new trend in other domains such as NLP and CVs with a focus on how to select informative training instances. It is important to shed some light in the context of how to better use the pre-training graphs data.

- The core idea the author proposed intuitively makes sense. Indeed, graphs with certain characteristics such as higher network entropy, larger density, higher average degree, higher degree variance, or a smaller scale-free exponent will contain a larger amount of information. Even though the authors do not provide theoretical justification, empirically, this work does provide good insight into GNN pre-training.

**Weaknesses**

- The intuition figure 1 is a bit hard to fully understand. For the first row, when scaling up the sample size, does it mean we keep the same number of pre-train graphs but we vary the same portion of sample size from each pre-train graph dataset? For the bottom row, the total number of pre-train graphs in the paper is eleven and the number of graphs in the figure is only up to ten. The author should provide what is the benchmark results if we use all the possible pre-train datasets to properly compare

- Directly using contrastive loss to define and measure predictive uncertainty is not questionable.

- The notation of this paper is a bit messy. For example, both scaler and vector use non-bolded lowercase (degree vector and node degree) which is not conventional.

- The selection mechanism is a bit complicated and heavily handcrafted. Currently, the graph selection policy uses a combination of 6 loss terms in total. Besides, loss terms work on a very different scale and are combined together with an additional time-adaptive parameter. I am not sure if this design is reasonable. Empirically, there is no proper ablation study on the choice of which graph property to be included in the pre-train graph selection criteria.

- There is an additional proximal regularization term in the model design, which aims to better preserve the knowledge or information contained in previous input data when we train on new incoming data, a similar design component in the continue learning paradigm. I am not able to fully follow the rationale of this design. The pre-training problem is completely different from the catastrophic forgetting setting since we will normally shuffle the training order of the samples after each epoch. The claim of "previous input data will be forgotten or covered by new incoming data" is invalid if we shuffle the data training order. Besides, the $\mathbf{\theta}$ parameters learned from the first $j$ graphs, does it mean in terms of the memory complexity of the model will be $j$ times larger since we need to store the previous training iteration of the model?

- Experimental results-wise, I have several concerns: 1) It seems the work is directly established on top of the GCC paper, but with completely different suits of pre-training datasets and downstream datasets. The choice of dataset section seems arbitrary and suspicious. Could the author directly work on the precious experiment setting in GCC? or what's the reason behind the complete switch of datasets?
 2) The baseline uses are quite outdated, using baselines only coming before the year 2020. A lot of recent work in terms of GNN pre-training or graph data augmentation should be included [1] - [8].  3) One suggestion to the authors, since the paper uses a lot of downstream tasks and lots of numbers in the result table, a better way to present the results can be considered. For example, providing an average rank number across all the datasets for each method will be informative and helpful to deliver the message to readers.  3) I will suggest including the full pre-training dataset results in the experiment table to see the effectiveness of the pre-train graph selection scheme. Or should I interpret the GCC results as the full pre-training dataset results?  4) Can the author provide the training time comparison? It seems that to fully

- Can the author comment on the relationship of this work to some recent adaptive graph positive sample generation work? It seems they all achieve a similar end goal by finding the proper training samples [3] [5] [6] [7].


[1] Hu, Ziniu, et al. "GPT-GNN: Generative pre-training of graph neural networks." KDD 2020.

[2] Xu, Dongkuan, et al. "Infogcl: Information-aware graph contrastive learning." NeurIPS 2021.

[3] Zhu, Yanqiao, et al. "Graph contrastive learning with adaptive augmentation." Proceedings of the Web Conference 2021. 2021.

[4] Zhu, Yanqiao, et al. "An Empirical Study of Graph Contrastive Learning." NeurIPS 2021.

[5] You, Yuning, et al. "Graph contrastive learning automated." ICML 2021.

[6] Lee N, Lee J, Park C. Augmentation-free self-supervised learning on graphs. In Proc. of the AAAI Conference on Artificial Intelligence 2022.

[7] Han et al. "G-Mixup: Graph Data Augmentation for Graph Classification." ICML 2022.

[8] Hou, Zhenyu, et al. "GraphMAE: Self-Supervised Masked Graph Autoencoders." KDD 2022.









**Summary Of The Paper:**

This paper studies an interesting question in terms of which graph datasets should be selected for GNN pre-training tasks. The authors propose a novel graph selector that is able to provide the most instructive data for the model. The criteria in the graph selector include predictive uncertainty and graph properties (graph entropy, density, degree, etc). Besides, they propose a data-active graph pre-training (APT) framework, which integrates the graph selector and the pre-training model into a unified framework.


**Summary Of The Review:**

Please refer to Strength And Weaknesses.

---

> ### Author Response · Authors · 2022-11-14
> **Response to Reviewer ajow [5/5]**
>
> **Regarding the measure of average rank number.**
>
> Thanks for this valuable suggestion. We have included the average rank in Tables 1 and 2 in the revised manuscript. The results of average rank also present our model's superiority.
>
> **Regarding the full pre-training dataset results.**
>
> sorry for the ambiguity. The GCC results are indeed the version of our model without the data selection scheme, which trains on full pre-training datasets. We have clarified this in Section 4.1 in the revised version.
>
>
> **Regarding the training time comparison.**
>
> The training time comparison of our model and the competitor GCC is included in Appendix G, under the title "Training time". The results show that the total training time of APT-L2 and APT is 18321.39 seconds and 18592.01 seconds respectively (including the time consumed in graph selection and regularization term), while the competitive graph pre-training model GCC takes 40161.68 seconds for the same number of training epochs on the same datasets. Besides, we also show that the time spent of graph selection and proximity regularization term only accounts for 3.87% and 0.45% of the total training time of APT. In the revised version, we have highlighted the training time analysis in the first paragraph in section 4 to avoid missing information.
>
>
> > Q7: Can the author comment on the relationship of this work to some recent adaptive graph positive sample generation work? It seems they all achieve a similar end goal by finding the proper training samples [3] [5] [6] [7].
>
> The mentioned papers [3, 5, 6] aim to construct positive pairs (i.e., typically generate/select augmented views of anchor samples) in self-supervised learning rather than selecting samples. In particular, they utilize ALL the samples (or uniformly sampled ones) rather than adaptively selecting some of them. We believe that the mentioned works belong to another line of research, and their contributions can also be embedded into our pre-training model, after the data selection process.
>
> The paper [7] proposed a mixup method to generate synthetic data based on a set of graphs and their class label. Such approach cannot be used to select data for pre-training, because (1) the pre-training data are unlabelled and (2) we aim to select less data rather than generate more data.
>
> **Reference:**
>
> [1] Ziniu Hu, Yuxiao Dong, Kuansan Wang, Kai-Wei Chang, and Yizhou Sun. 2020. GPT-GNN: Generative pre-training of graph neural networks.In SIGKDD, pages 1857–1867.
>
> [2] Dongkuan Xu, Wei Cheng, Dongsheng Luo, Haifeng Chen, and Xiang Zhang.  2021. Infogcl: Information-aware graph contrastive learning, pages 30414–30425.
>
> [3] Zhu, Yanqiao, Yichen Xu, Feng Yu, Q. Liu, Shu Wu and Liang Wang. 2021. Graph contrastive learning with adaptive augmentation. In WWW, pages 2069-2080.
>
> [4] Yanqiao Zhu, Yichen Xu, Qiang Liu, and Shu Wu. An empirical study of graph contrastive learning. 2021. In NeurIPS D&B.
>
> [5] Yuning You, Tianlong Chen, Yang Shen, and Zhangyang Wang. 2021. Graph contrastive learning automated. In ICML, pages 12121–12132.
>
> [6] Namkyeong Lee, Junseok Lee, and Chanyoung Park. 2022. Augmentation-free self-supervised learning on graphs. In AAAI, pages 7372–7380.
>
> [7] Xiaotian Han, Zhimeng Jiang, Ninghao Liu, and Xia Hu. 2022. G-mixup: Graph data augmentation for graph classification. In ICML, pages 8230–8248.
>
> [8] Zhenyu Hou, Xiao Liu, Yukuo Cen, Yuxiao Dong, Hongxia Yang, Chunjie Wang, and Jie Tang. 2022. Graphmae: Self-supervised masked graph autoencoders. In SIGKDD, pages 594–604.
>
> [9] Edgar Simo-Serra, Eduard Trulls, Luis Ferraz, Iasonas Kokkinos, and Francesc Moreno-Noguer. 2014. Fracking deep convolutional image descriptors. arXiv preprint arXiv:1412.6537.
>
> [10] Abhinav Shrivastava, Abhinav Gupta, and Ross Girshick. 2016. Training region-based object detectors with online hard example mining. In CVPR, pages. 761–769.
>
> [11] Suh, Yumin, Bohyung Han, Wonsik Kim and Kyoung Mu Lee. 2019. Stochastic Class-Based Hard Example Mining for Deep Metric Learning. In CVPR, pages 7244-7252.
>
> [12] Florian Schroff, Dmitry Kalenichenko, and James Philbin. 2015. FaceNet: A unified embedding for face recognition and clustering. In CVPR, pages 815-823.
>
> [13] Kirkpatrick, James, Razvan Pascanu, Neil C. Rabinowitz, Joel Veness, Guillaume Desjardins, Andrei A. Rusu, Kieran Milan, John Quan, Tiago Ramalho, Agnieszka Grabska-Barwinska, Demis Hassabis, Claudia Clopath, Dharshan Kumaran and Raia Hadsell. 2017. Overcoming catastrophic forgetting in neural networks. In Proceedings of the national academy of sciences, pages 3521-3526.
>
> [14] Hu, Weihua, Matthias Fey, Marinka Zitnik, Yuxiao Dong, Hongyu Ren, Bowen Liu, Michele Catasta and Jure Leskovec.  2020. Open graph benchmark: Datasets for machine learning on graphs. In NeurIPS, pages 22118-22133.

---

> ### Author Response · Authors · 2022-11-14
> **Response to Reviewer ajow [4/5]**
>
>
> **Regarding the GNN pre-training or graph data augmentation baselines.**
>
> Thanks for bringing these works into our attention. Additional baseline [5] is included in Tables 1 and 2, and our model still outperforms the rest in most cases. Below we discuss the experimental details in using [5], and explain why the other mentioned works are difficult to adapt in our experimental setting. The discussion of these related works is also included in the revised manuscript, see Section 5.
>
> 1. In JOAO [5], the input graphs are assumed to be in the same domain (i.e., biochemistry) and the downstream task is limited to graph classification. Adaptation to our cross-domain setting is relatively straightforward (with graph classification as downstream task). For the node classification task, we simply input the RWR subgraphs from GCC to the presented models in [5], and treat the learned subgraph representation as node representation in their setting. The experimental results are presented in the following tables, which still show our superiority.
>
> | Dataset|  brazil | dd242| dd68|dd687|wisconsin|cornell|cora | pubmed         | ogbarxiv|ogbproteins|
> |---------|:----:|:-----:|:-----:|:------:|:-----:|:------:|:-----:|:------:|:------:|:-----:|
> | JOAO (freeze)    |         71.22(7.21)        |         7.98(2.90)         |         12.36(2.59)        |         5.34(1.43)         |         42.69(8.15)        |        43.16(5.67)       |       18.13(2.82)      |       41.05(0.87)      |       /          |           /          |
> | APT (freeze)    |   73.39(2.55)   |        16.57(0.94)       |        12.08(0.89)       |   10.35(1.24)       |        53.38(1.19)       |       47.37(1.29)      | 36.69(0.49) | 46.88(0.21) | 22.04(0.29) | 62.29(0.55) |
> | JOAO (rand, fine-tune)   |         72.14(6.74)        |         10.93(2.85)        |         8.08(2.15)         |         7.40(3.48)         |        45.38(13.30)        |       45.26(10.31)       |       29.93(2.84)      |       42.01(0.68)      |   /  |     /          |
> | JOAO (fine-tune)     |         75.00(5.76)        |         10.54(3.07)        |         7.56(1.94)         |         8.77(2.39)         |         50.0(12.28)        |      42.11(10.26)      |       29.34(3.04)      |       42.21(0.88)      |   /   |   /     |
> | APT (fine-tune)      | 79.67(2.30) | 28.62(0.55) | 20.30(1.13) | 12.80(1.54) | 67.08(1.75) |       52.15(2.25)      | 47.51(0.62) | 51.30(0.16) | 27.40(4.87) | 61.64(0.35) |
>
>
> Table: Micro F1 scores of GCC and JOAO[5] in the node classification task.
>
> | Dataset   |  imdb-binary  |  dd    |   msrc-21  |
> |-----------|:--------------:|:-------:|:--------:|
> | JOAO (freeze)   |   63.90(3.48) |55.97(3.61)|  5.09(2.65)     |
> | APT (freeze)    | 73.00(0.50) |75.83(0.31)  |  13.81(1.06)  |
> | JOAO (rand, fine-tune) |67.70(3.35)|62.10(4.31)|11.40(3.06)|
> | JOAO (fine-tune) | 68.50(3.61) | 62.61(4.99) | 10.18(1.72)   |
> | APT (fine-tune) |  76.27(1.20) | 75.69(1.42) | 24.41(1.82)   |
>
>
> Table: Micro F1 scores of GCC and JOAO[5] in the graph classification task.
>
> 2. GPT-GNN [1] can only be finetuned on graphs of the same type as the pre-training one, including node type, edge type and node attributes. GraphMAE [8] designs a self-supervised task based on node attribute reconstruction, which requires the dimension and meaning of node attributes to be consistent. Therefore, it is difficult to adapt these two works in our experiments since the datasets are from various domains.
>
> 3. The graph contrastive learning methods [2, 3, 6, 7] have to be trained and tested on the same graph dataset since they utilize graph-specific signals (e.g., node attributes or labels). Thus adaptation in cross-domain datasets is not straightforward.
>
> 4. The benchmark paper [4] provides empirical evaluations of critical designs in existing graph contrastive learning. Among the methods discussed in [4], only GCC and GraphCL fit our experimental setting, and we have already included them as the baselines.

---

> ### Author Response · Authors · 2022-11-14
> **Response to Reviewer ajow [3/5]**
>
> > Q6: Experimental results-wise concerns. 1) It seems the work is directly established on top of the GCC paper, but with completely different suits of pre-training datasets and downstream datasets. The choice of dataset section seems arbitrary and suspicious. Could the author directly work on the precious experiment setting in GCC? or what's the reason behind the complete switch of datasets? 2) The baseline uses are quite outdated, using baselines only coming before the year 2020. A lot of recent work in terms of GNN pre-training or graph data augmentation should be included [1] - [8]. 3) One suggestion to the authors, since the paper uses a lot of downstream tasks and lots of numbers in the result table, a better way to present the results can be considered. For example, providing an average rank number across all the datasets for each method will be informative and helpful to deliver the message to readers. 4) I will suggest including the full pre-training dataset results in the experiment table to see the effectiveness of the pre-train graph selection scheme. Or should I interpret the GCC results as the full pre-training dataset results? 5) Can the author provide the training time comparison? It seems that to fully
>
> **Regarding the choice of pre-training data and test data.**
>
> Sorry for the unclear description. We here first present the experimental results under the experiment setting of GCC as requested. Then, we illustrate the criterion of selecting the pre-training data and test data in our paper.
>
> 1. The following table shows the performance of our model and the strongest competitor GCC, under the experiment setting of GCC. (The performance of GCC is directly taken from its paper.) The results indicate that our model still outperforms GCC under the experimental setting of GCC.
>
>
> | Dataset|  US-Airport |   H-index   |
> |------------|:-----------:|:-----------:|
> | GCC |     67.2    |     80.6    |
> | Ours | 70.50(6.08) | 82.28(1.48) |
>
> Table: Micro F1 scores of GCC and our model in the node classification task in fine-tuning mode, under the experimental setting of GCC.
>
>
> | Dataset|     IMDB-B   |    IMDB-M   |    COLLAB   |    RDT-B    |    RDT-M    |
> |--------|:-----------:|:-----------:|:-----------:|:-----------:|:-----------:|
> | GCC |  73.8  |  50.3    |   81.1  |  87.6 |     53.0    |
> | Ours  |  76.27(1.20) | 50.50(1.08) | 81.23(0.86) | 92.20(2.43) | 53.28(2.33) |
>
> Table: Micro F1 scores of GCC and our model in the graph classification task in fine-tuning mode, under the experimental setting of GCC.
>
> 2. We then illustrate the criterion of selecting the pre-training data and test data. Overall, the selected pre-training data and test data should cover a wide spectrum of domains. The consideration of the graphs for pre-training and test is as follows. When selecting pre-training data, we hope that the graph size is at least hundreds of thousands to contain enough information for pre-training. When selecting test data, we hope that: (1) some test data is in the same domain as the pre-training data, and some is cross-domain, so as to comprehensively evaluate our model’s in and across-domain transferability. Accordingly, the in-domain test data is selected from the type of movie and citations, and the others test data are across-domain; (2) the size of test graphs can scale from hundreds to millions. The selection criterion have been included in Appendix C in the revised version.

---

> ### Author Response · Authors · 2022-11-14
> **Response to Reviewer ajow [2/5]**
>
> > Q4: The selection mechanism is a bit complicated and heavily handcrafted. Currently, the graph selection policy uses a combination of 6 loss terms in total. Besides, loss terms work on a very different scale and are combined together with an additional time-adaptive parameter. I am not sure if this design is reasonable. Empirically, there is no proper ablation study on the choice of which graph property to be included in the pre-train graph selection criteria.
>
> The graph selection mechanism is based on both the predictive uncertainty and the carefully selected graph properties. The choice of these measures is intuitively explained in Section 3.1, and more detailed analysis of the graph properties is provided in Appendix A. Moreover, this choice is empirically justified, and below we discuss it in more detail.
>
> 1. Each loss term is the z-score normalized value, and thus insensitive to the scale of graph properties and graph-level predictive uncertainty.
>
> 2. The time-adaptive parameter is used to balance between predictive uncertainty and graph properties. Detailed explanation can be found on page 6 of the manuscript. The current design of time-adaptive parameter is empirically the best among other alternatives. More experiment results are provided in Appendix G, under the title ``The choice of \beta_t , its alternatives, and ablation study''.
>
> 3. Experimental results in Figure 7 of Appendix E show that the selected graph properties all exhibit strong correlations with the performance of the pre-trained model. On the contrary, some other commonly-used properties (including clique number, transitivity, degree assortativity and average clustering coefficient) exhibit little or no correlation with the performance. Properties like diameter and Wiener index are not considered here due to their high computational complexity. Overall, extensive experiment results verify the use of the current five graph properties, and also show the ineffectiveness of the others.
>
> 4. More ablation studies can be found in Table 10 and Table 11 in Appendix G. We present the results when only one graph property is utilized in our model. We find that the five graph properties used in our model are all indispensable, and the most important one probably varies for different tasks and datasets.
>
> Last, we mention that the selection mechanism of graph properties is flexible. Other properties could be easily embedded in our model.
>
>
> > Q5: There is an additional proximal regularization term in the model design, which aims to better preserve the knowledge or information contained in previous input data when we train on new incoming data, a similar design component in the continue learning paradigm. I am not able to fully follow the rationale of this design. The pre-training problem is completely different from the catastrophic forgetting setting since we will normally shuffle the training order of the samples after each epoch. The claim of "previous input data will be forgotten or covered by new incoming data" is invalid if we shuffle the data training order. Besides, the θ parameters learned from the first j graphs, does it mean in terms of the memory complexity of the model will be j times larger since we need to store the previous training iteration of the model?
>
> **Regarding the pre-training process.**
>
> Sorry for the misunderstanding. In our proposed graph pre-training pipeline, instead of swallowing all the pre-training graphs as a whole and then shuffling their training order, the pre-training model takes the pre-training graphs one by one in a sequential order and enhances itself in a progressive manner. This is because in different stages of pre-training, the least certain data points for the current pre-training model would be different. The pre-training model thus first properly learns the samples from one graph, and then learns from another graph that it has least knowledge of. By doing this, the pre-training model is able to reinforce itself on highly uncertain data in next training iterations.
>
> **Regarding the θ parameters.**
>
> Sorry for the inaccurate description. Here we follow the setting in [13]. The proximal regularization only takes into consideration the model parameters trained on the previous graph. Besides, the regularization term is applied on the first three layers of the pre-training model. So the memory cost is at most 2 times larger than the pre-training model. In the experiment, the total number of parameters in the pre-training model is 190,544, in the same order of magnitude as classical GNNs (GraphSAGE, GraphSAINT, etc.). This number is relatively small among models in open graph benchmark (OGB) [14]. Therefore, the memory requirement would not be a bottleneck.

---

> ### Author Response · Authors · 2022-11-14
> **Response to Reviewer ajow [1/5]**
>
> Dear reviewer ajow,
>
> We really appreciate your valuable comments. We have followed closely the suggestions, and made clarifications and revisions accordingly. We hope the newly provided content could help to further strengthen our work.
>
> > Q1: The intuition figure 1 is a bit hard to fully understand. For the first row, when scaling up the sample size, does it mean we keep the same number of pre-train graphs but we vary the same portion of sample size from each pre-train graph dataset? For the bottom row, the total number of pre-train graphs in the paper is eleven and the number of graphs in the figure is only up to ten. The author should provide what is the benchmark results if we use all the possible pre-train datasets to properly compare.
>
> Sorry for the unclear description. In what follows, we clarify the questions in Figure 1 one-by-one.
>
> **Regarding the first row of Figure 1.**
>
> When scaling up the sample size, the graphs used for pre-training are kept as all eleven pre-training graphs. For a fixed sample size, the samples are directly taken from the backbone pre-training model according to its sampling strategy. Conventionally, the number of samples from different pre-training graphs is proportional to the number of nodes of pre-training graphs. We have added detailed descriptions in Figure 1' caption in the revised version.
>
> **Regarding the maximal number of graphs in the bottom row of Figure 1.**
>
> For a fixed number of pre-training graphs, the bottom row shows the results (mean and std) of different combinations of input graphs. When the entire eleven input graphs are used, we only have one possible combination. Thus we did not plot it in Figure 1.
>
> **Regarding benchmark results of using all possible pre-training graphs.**
>
> The results of using all eleven datasets are already in Tables 1 and 2, under the label "GCC (freeze)". In the revised manuscript we also add this to Figure 1 as requested. The conclusion remains that adding input graphs does not improve and sometimes even deteriorates the generalization of the pre-trained model.
>
> > Q2: Directly using contrastive loss to define and measure predictive uncertainty is not questionable.
>
> We are not sure whether we understand this question correctly. We provide two explanations of using contrastive loss as the measure of predictive uncertainty and select data according to it. First, the contrastive loss is exactly the objective function of our model. So what we do is actually select the samples with the greatest contributions to the objective function (i.e., select the samples with the greatest InfoNCE loss). Such strategy is justified to be an effective way to select samples for training in several works of hard example mining [9-10], and could accelerate convergence and enhance the discriminative power of the learned representations [11-12]. This further adds to the integrity and rationality of our strategy. Third, as the loss function of our model, the contrastive loss is already computed during the training, and thus no additional computation expense is needed in the data selection phase.
> The explanations have been included in Appendix H in the revised version. Beyond these explanations, could you remind us if there remains anything unclear descriptions. We are happy to make any further revisions.
>
> > Q3: The notation of this paper is a bit messy. For example, both scaler and vector use non-bolded lowercase (degree vector and node degree) which is not conventional.
>
> Thanks for pointing it out. We have made revisions by using bolded lowercase and non-bolded lowercase to represent all the vectors and scalars, respectively.

---

> ### Author Response · Authors · 2022-11-24
> **Any unanswered questions yet?**
>
> Dear Reviewer ajow,
>
> Thanks again for your detailed review and questions. We have provided detailed answers to them in the rebuttal and would like to kindly ask if you still have any unanswered questions about our paper?
>
> Best,
> Authors

---

> ### Comment · Area_Chair_raJQ · 2022-12-13
> **Notice**
>
> Dear Reviewer,
>
> If you do not want to update your score, please at least acknowledge that you have already read the rebuttal.

---

### Comment · Area_Chair_raJQ · 2022-11-22
**Please respond as soon as possible if you still have questions on the paper.**

Please respond as soon as possible if you still have questions on the paper.

---

> ### Comment · Area_Chair_raJQ · 2022-11-29
> **Please respond to the authors by Nov. 30**
>
> Please indicate whether the authors' rebuttal addresses your concerns.
>
> If you still have questions, please ask as soon as possible.

---

> > ### Comment · Area_Chair_raJQ · 2022-12-05
> > **Zoom Meeting**
> >
> > For all reviewers, which have not responded to the authors, I will have to ask you to meet via Zoom. If you want to avoid such an additional step, please respond by Dec. 5.

---

### Decision · Program_Chairs · 2023-01-20

**Decision:**

Reject

**Justification For Why Not Higher Score:**

NA

**Justification For Why Not Lower Score:**

NA

**Metareview: Summary, Strengths And Weaknesses:**

This paper investigates how to choose the most effective graph datasets for pre-training Graph Neural Network (GNN) models. The authors propose a graph selector that utilizes predictive uncertainty and various graph properties, such as graph entropy and density, to select the most instructive data. They also present a data-active graph pre-training (APT) framework that combines the graph selector and the pre-training model into a single system.

The reviewers raised two major concerns on the effectiveness of the proposed pre-training approach for Graph Neural Networks (GNNs) and the practicality of the heavily handcrafted training objective: (1) The authors did not provide evidence of the benefits of the pre-training approach under the proper experimental setting, in which the encoder is frozen during fine-tuning for downstream tasks. (2) The handcrafted training objective heavily replies on certain graph properties，while the downstream graphs may have very different beneficial graph statistics. Therefore, the pre-training can be less effective without extensive tuning.

**Summary Of Ac-Reviewer Meeting:**

NA